# Vegetation enhances curvature-driven dynamics in meandering rivers

Alvise Finotello [1] ✉, Alessandro Ielpi [2], Mathieu G. A. Lapôtre [3], Eli D. Lazarus [4], Massimiliano Ghinassi[1], Luca Carniello [5], Serena Favaro[5], Davide Tognin [5] & Andrea D'Alpaos [1]

Stabilization of riverbanks by vegetation has long been considered necessary to sustain single-thread meandering rivers. However, observation of active meandering in modern barren landscapes challenges this assumption. Here, we investigate a globally distributed set of modern meandering rivers with varying riparian vegetation densities, using satellite imagery and statistical analyses of meander-form descriptors and migration rates. We show that vegetation enhances the coefficient of proportionality between channel curvature and migration rates at low curvatures, and that this effect wanes in curvier channels irrespective of vegetation density. By stabilizing low-curvature reaches and allowing meanders to gain sinuosity as channels migrate laterally, vegetation quantifiably affects river morphodynamics. Any causality between denser vegetation and higher meander sinuosity, however, cannot be inferred owing to more frequent avulsions in modern non-vegetated environments. By illustrating how vegetation affects channel mobility and floodplain reworking, our findings have implications for assessing carbon stocks and fluxes in river floodplains.

Meandering rivers are widespread in fluvial lowlands and represent one of the most dynamic landforms on Earth[1,2]. Besides being of critical importance for geological and engineering applications[3,4], the dynamics of river meandering also exert primary controls on the biogeochemical cycles that make Earth's climate suitable for life[5–8].

The formation and development of meandering rivers have long been considered intimately linked to the stabilizing influence of vegetation on riverbanks. This includes bank strengthening provided by plant roots, as well as vegetation-enhanced generation and retention of pedogenic muds[9–11]. This hypothesis has been corroborated by physical experiments where vegetation facilitated the transition from multi- to single-thread rivers[12,13]. Moreover, the appearance of known indicators of river meandering in the geologic record has been observed to broadly coincide with the emergence of plant life on Earth *ca* 440 million years ago. Specifically, mud-rich sedimentary deposits characterized by laterally-accreting inclined heterolithic stratifications, which have classically been considered diagnostic of meander point bars, are uncommon prior to the evolution of vascular plants[9,10,14]. This observation led to the hypothesis that vegetation was instrumental to the spread of meandering rivers in the Paleozoic[15–18].

However, recent empirical evidence[19–22] and physics-based modeling[23] indicate that meandering rivers can also develop in landscapes devoid of plant life—including on Mars, where records of ancient vegetation are missing[23,24]. Unvegetated meandering river systems have now been recognized on several continents, displaying a diverse set of geomorphic units akin to those observed in vegetated alluvial floodplains, including subtle channel levees, crevasse splays, and point bars[5,15,20,21,25–29]. These observations suggest that single-thread meandering rivers can be sustained even in the absence of land plants, given favorable conditions of channel slope and as long as

[1]Department of Geosciences, University of Padua, Padua, Italy. [2]Earth, Environmental and Geographic Sciences, University of British Columbia, Okanagan Campus, Kelowna, BC, Canada. [3]Department of Earth and Planetary Sciences, Stanford University, Stanford, CA, USA. [4]Department of Geography and Environmental Science, University of Southampton, Southampton, UK. [5]Dept. of Civil, Environmental, and Architectural Engineering, University of Padua, Padua, Italy. ✉e-mail: alvise.finotello@unipd.it

other cohesive agents, such as mud or ice, provide the necessary bank strength[5,26,30]. Yet, these observations also prompt new questions regarding the extent to which vegetation, or lack thereof, affects river meander morphology and dynamics, both directly and indirectly[28,31,32].

Here, we analyze a globally distributed sample of modern meandering rivers spanning a range of riparian vegetation densities, with the goal of elucidating how vegetation influences the planform shape and evolution of meanders. We first demonstrate how denser riparian vegetation correlates with enhanced meander sinuosity, skewness, and curvature, although planform controls by vegetation are rather subtle and cannot be completely isolated from the effects of high aridity and flashy hydrological regimes. This is especially relevant in modern unvegetated rivers, where meander growth and evolution are prematurely disrupted by meander chute cutoffs and frequent avulsions. Nonetheless, we confirm that vegetation demonstrably reduces the rate of river lateral migration, and enhances the coefficient of proportionality between migration rate and channel curvature. This enhancement is most evident at relatively low curvature and wanes in curvier channels regardless of vegetation density. Irrespective of aridity, vegetation effectively stabilizes low curvature reaches and allows meanders to gain sinuosity as they migrate laterally, thereby leaving measurable imprints on the morphodynamics of meandering rivers.

## Results

Our dataset consists of 54 single-thread meandering rivers found in distinct biomes worldwide (Fig. 1a). We quantified the planform morphology of individual reaches by manually digitizing riverbanks from aerial and satellite images, and estimated the density of riparian vegetation from multi-annual statistics of the remotely sensed Normalized Difference Vegetation Index (NDVI) (Fig. 1a and Methods). We defined three classes of rivers based on the computed NDVI: unvegetated (NDVI ≤ 0.2; $n_{unveg}$ = 16), semi-vegetated (0.2 < NDVI ≤ 0.4; $n_{semiveg}$ = 11), and vegetated (NDVI > 0.4; $n_{veg}$ = 27) (Fig. 1a, b). For each class, we analyzed the width-adjusted morphometric properties of meander planforms at both the scale of single meander bends and individual meandering river reaches, using classical uni- and multivariate statistical methods in the river-meander literature[33–36] (Fig. 1a and Methods). Finally, to unravel the role of vegetation in driving meander-planform evolution, we integrated measurements of river curvature ($C^*$ [m$^{-1}$]) and lateral migration rates ($M_R^*$ [m/yr]) derived from dynamic time-warping (DTW) analyses[37,38] (Fig. 1d) applied to a subset of actively migrating rivers ($n$ = 32, Fig. 1a) where pairs of aerial images were available at sufficient pixel resolution to reconstruct the position of channel centerlines over multiannual timespans (Methods).

### Effects of vegetation on meander planforms

Scaling relationships between fundamental morphometric characteristics of meander planforms (Fig. 2) are remarkably similar among unvegetated, semi-vegetated, and vegetated rivers (Fig. 2a–d). We find that meander morphometric features—namely Cartesian ($L^*$ [m]) and intrinsic ($\ell^*$ [m]) wavelengths, meander amplitude ($A^*$ [m]), and radius of curvature ($R^*$ [m])—all exhibit statistically significant ($p < 0.01$) power-law relationships of the form $z^* = a \cdot (B^*)^b$, where $z^*$ is a given meander morphometric variable and $B^*$ ([m]) represents the channel width averaged along individual meander bends (Fig. 2a–d). Notably, whereas power-law prefactors ($a$) differ slightly among distinct vegetation classes, power-law exponents ($b$) closely approximating 1 imply linear scaling

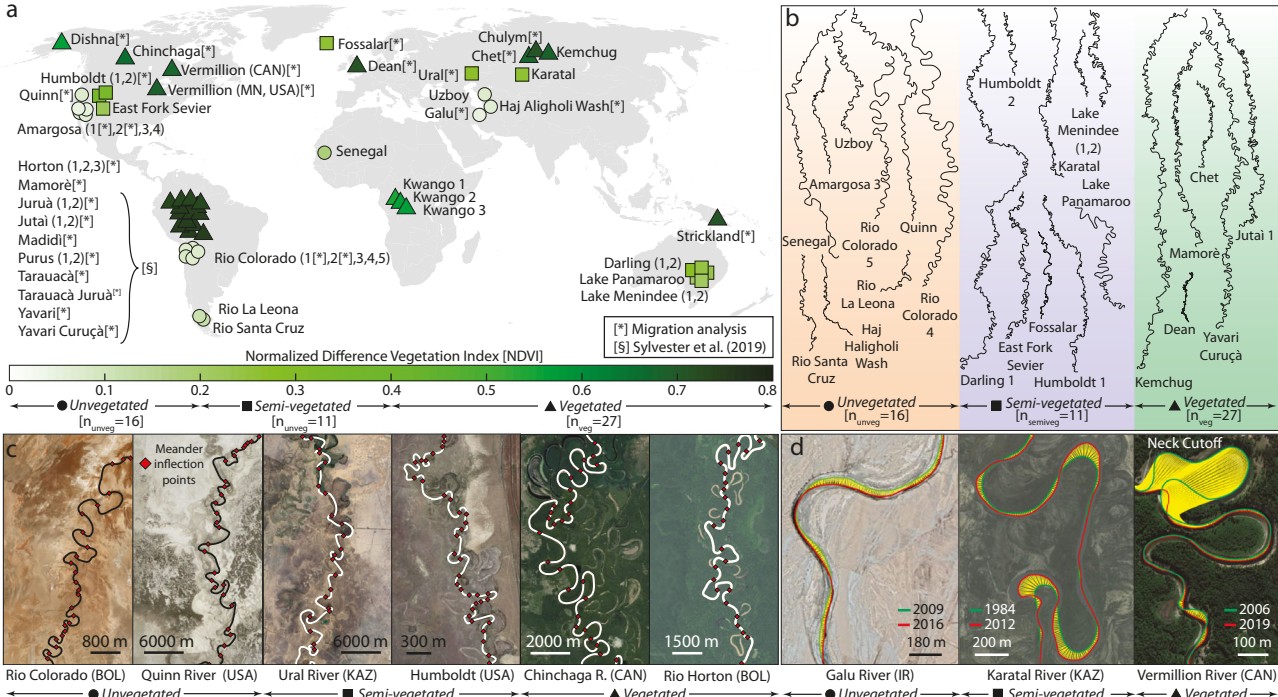

Fig. 1 | **Overview of the studied rivers and quantification of meander morphologies and dynamics. a** Global map with locations of selected rivers analyzed in the present study. Colors denote different Normalized Difference Vegetation Index (NDVI) computed from Landsat 7 Collection 1 Tier 1 8-Day NDVI composites. Symbols represent the separation of our dataset into three distinct vegetation classes according to NDVI. The symbol "[*]" indicates a subset of rivers for which migration analysis was carried out, whereas the symbol "[§]" shows data collection provided by Sylvester et al.[37]. **b** Planform morphology of selected subsets of unvegetated, semi-vegetated, and vegetated meandering rivers. Each meander in the displayed channel traces is planimetrically scaled with its average half-width. Scale is arbitrary. **c** Examples of meander-bend segmentation for rivers found in distinct vegetational contexts. The river centerline is highlighted, together with the position of meander inflection points (red dots) computed based on the channel axis curvature (see Methods). **d** Examples of meandering-river migration trajectories, in yellow, computed through dynamic time warping (DTW) applied to pairs of consecutive river centerlines (see Methods).

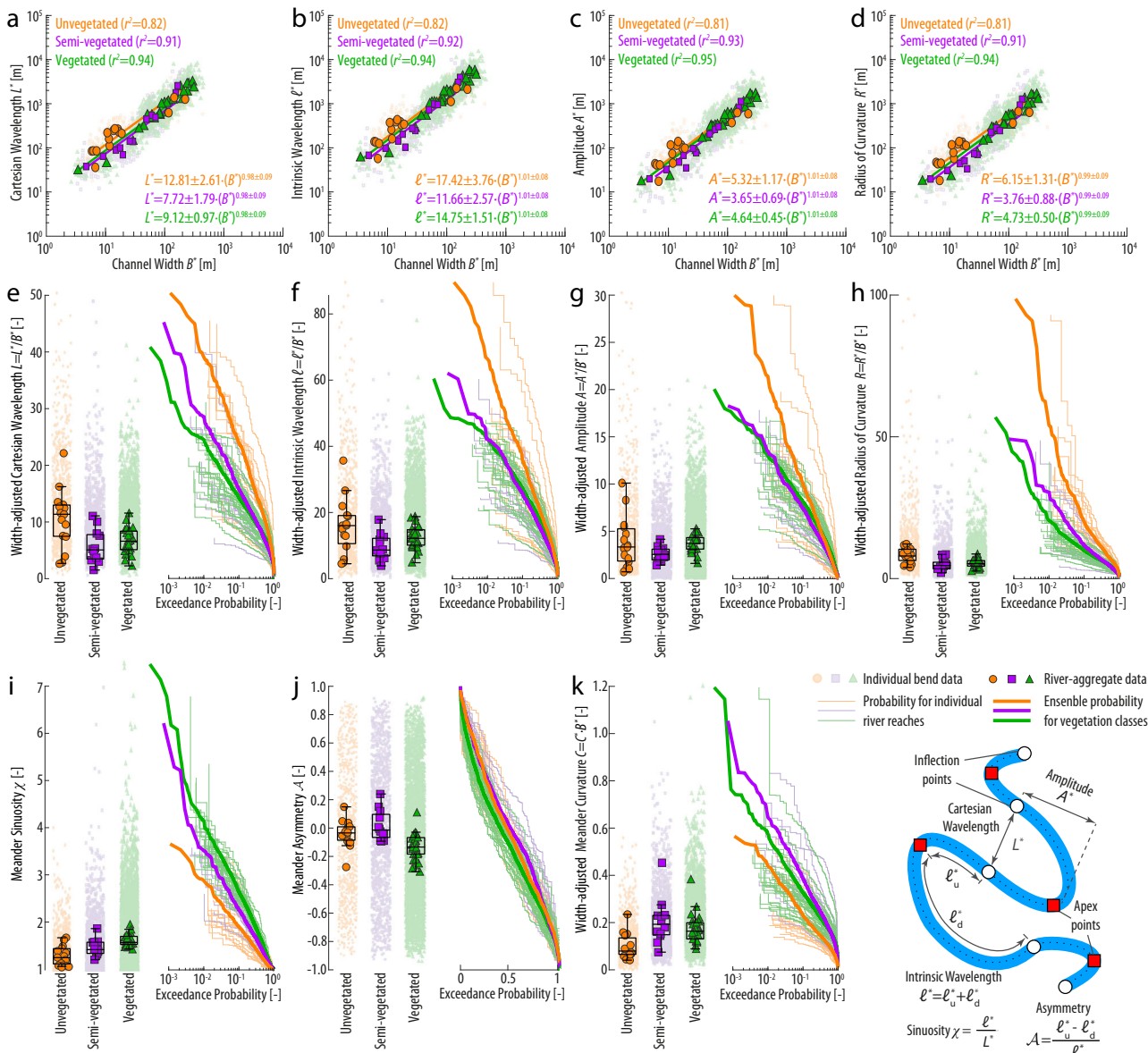

**Fig. 2 | Planform morphometrics of river meanders as a function of riparian vegetation density. a–d** Meander Cartesian wavelength ($L^*$), intrinsic wavelength ($\ell^*$), amplitude ($A$), and radius of curvature ($R^*$) are plotted against the channel width averaged along individual meander bends ($B^*$) for unvegetated, semi-vegetated, and vegetated river data separately. Reduced-size shaded markers in the background represent data of individual meander bends (only 50% of data points are plotted to improve readability), whereas larger markers denote aggregated data for each river in our dataset. Bends characterized by sinuosity $\chi < 1.2$ are filtered out. Continuous lines represent the power-law fit of aggregated data points obtained through linear regressions on log-transformed data. Power-law regression coefficients ($a$ and $b$) and R-squared coefficients (denoted as $r^2$ to avoid confusion with the meander radius) are also reported in each panel for individual vegetation classes. **e–k** Boxplots and empirical probability distributions, plotted as exceedance probability, of meander width-adjusted cartesian wavelength ($L = L^*/B^*$), width-adjusted intrinsic wavelength ($\ell = \ell^*/B^*$), width-adjusted amplitude ($A = A^*/B^*$), width-adjusted radius of curvature ($R = R^*/B^*$), sinuosity ($\chi$), asymmetry ($\mathscr{A}$), and width-adjusted curvature ($C = C^* \cdot B^*$). In each boxplot, reduced-size shaded markers in the background represent data of individual meander bends, whereas larger markers denote aggregated data for each river in our dataset. The median of river-aggregated data is denoted by the central mark, while the lower and upper boundaries of the box represent the 25th and 75th percentiles, respectively. The whiskers reach out to encompass the most extreme data points that are not identified as outliers. In exceedance probability plots, thin lines denote the distribution of individual rivers, whereas thicker lines represent the ensemble probability for unvegetated, semi-vegetated, and vegetated rivers. The inset in the lower-right corner shows the extraction of individual morphometric variables based on a synthetic meandering river reach.

relationships between $B^*$ and all analyzed morphometric descriptors. Thus, to compare meandering rivers of different sizes, we investigate the distributions of dimensionless, width-adjusted meander wavelengths ($L = L^*/B^*$; $\ell = \ell^*/B^*$), amplitude ($A = A^*/B^*$), radius of curvature ($R = R^*/B^*$), and curvature ($C = C^* \cdot B^*$), as well as the distributions of width-independent meander sinuosity ($\chi = L^*/\ell^*$) and asymmetry ($\mathscr{A}$) (Fig. 2e–k). The latter parameter is computed as $\mathscr{A} = (\ell^*_u - \ell^*_d)/\ell^*$, where $\ell^*_u$ and $\ell^*_d$ denote the distance between the meander

apex and its upstream and downstream endpoints (Fig. 2, Inset), respectively, so that negative (positive) values of $\mathscr{A}$ indicate meanders skewed in the upstream (downstream) direction.

Two-sample Wilcoxon Rank Sum (WRS) and Kolmogorov-Smirnov (KS) tests, performed at a standard 5% significance level on river-aggregated data, highlight statistically significant differences in morphometric relationships between vegetated rivers (used as the control group) and both unvegetated and semi-vegetated

rivers. The KS tests all reject the null hypothesis that the morpho-metrics of unvegetated and semi-vegetated rivers come from the same distribution as those of vegetated rivers. Additionally, the WRS tests reject the null hypothesis that the morphometric parameters of both unvegetated and semi-vegetated rivers have the same median as vegetated rivers in all cases except for the meander amplitude in unvegetated rivers and meander wavelengths and curvature in semi-vegetated rivers.

Compared to their vegetated counterparts, rivers in unvegetated settings typically feature longer meander wavelengths and larger radii of curvature relative to channel width, with lower meander sinuosity and width-adjusted curvature (Fig. 2e, f, h, i, k). Specifically, whereas in vegetated rivers 26% of meanders have a sinuosity $\chi > 2$ and 13% have a curvature $C > 0.3$, these percentages decrease to 7% and 5%, respectively, for unvegetated rivers. Vegetation also appears to correlate with bend asymmetry: meanders in vegetated and semi-vegetated rivers are more strongly upstream-skewed (i.e., lower values of $\mathscr{A}$) whereas unvegetated rivers host more symmetric bends on average (Fig. 2j). Whereas these differences emerge when data are treated as ensemble averages (i.e., binned over the entire data set), distributions associated with individual rivers display a wide dispersion (Fig. 2e–k). Thus, meander morphologies overlap considerably among different vegetation classes.

Because the observed differences are subtle and yet systematic, correlations between planform expression and vegetation density might emerge more clearly from multivariate analyses of meander morphometries. Following a tested approach in river morphodynamics[33–35,39], we applied Principal Component Analysis (PCA) (Supplementary Method 1) to a set of distinct morphometric variables derived from river-aggregated distributions of meander planform features (Table 1 and Methods). As demonstrated by Howard and Hemberger[35], morphological variations among freely meandering streams can be resolved by morphometric parameters related to meander wavelength, skewness, sinuosity, and curvature. Hence, we considered the first- to fourth-order statistics of the probability distributions of meander sinuosity ($\chi$), asymmetry ($\mathscr{A}$), and both width-adjusted curvature ($C$) and intrinsic wavelength ($\ell$).

All variables were computed for both half and full meander bends (Table 1 and see Methods), except for $C$, for which we considered the reach-averaged values. Whereas the resulting dataset consists of 28 variables overall, we performed PCA on a subset of 17 variables (Fig. 3) that succinctly captures the variation in the original data while avoiding redundancy from cross-correlations. The first three principal components (i.e., PCs $a_1$, $a_2$, and $a_3$) account for about 55% of the total variance in the dataset, with the relative importance of higher-order PCs decaying exponentially (Supplementary Fig. 1). Data from unvegetated and vegetated meandering rivers cluster in different parts of PC biplots, especially in the ($a_1$; $a_2$) and ($a_1$; $a_3$) spaces where unvegetated rivers cluster in the negative $a_1$ half-plane opposite to the vegetated data (Fig. 3a–c; Supplementary Method 1). Data for semi-vegetated rivers, in contrast, do not form a well-clustered group and generally fall somewhat in between the unvegetated and vegetated data clusters (Fig. 3a–c). These results hold even when different ensembles of morphometric variables are considered (Supplementary Figs. 2 and 3), and altogether suggest that data separation is predominantly driven by higher meander sinuosity, curvature, and degree of upstream skewing (i.e., lower $\mathscr{A}$) in vegetated rivers compared to their unvegetated and semi-vegetated counterparts. Higher $\chi$, larger $C$, and lower $\mathscr{A}$ values are typically associated with late-stage meander growth[40,41], suggesting that denser riparian vegetation correlates with the continued growth of meanders until they reach morphodynamic maturity. At this stage, bends become highly sinuous and are more likely to shortcut themselves through neck cutoff[40], which marks the endpoint of a meander evolution (Fig. 1d).

**Table 1 | List of morphometric variables used to characterize meandering rivers**

| Morphometric variable | Symbol | Description |
|---|---|---|
| Sinuosity | $\chi_{av_h}$ | Mean half-meander sinuosity |
| | $\chi_{av_f}$ | Mean full-meander sinuosity |
| | $\chi_{va_h}$ | Variance of half-meander sinuosity |
| | $\chi_{va_f}$ | Variance of full-meander sinuosity |
| | $\chi_{sk_h}$ | Skewness of half-meander sinuosity |
| | $\chi_{sk_f}$ | Skewness of full-meander sinuosity |
| | $\chi_{kr_h}$ | Kurtosis of half-meander sinuosity |
| | $\chi_{kr_f}$ | Kurtosis of full-meander sinuosity |
| Intrinsic wavelength | $\ell_{av_h}$ | Mean half-meander intrinsic wavelength |
| | $\ell_{av_f}$ | Mean full-meander intrinsic wavelength |
| | $\ell_{va_h}$ | Variance of half-meander intrinsic wavelength |
| | $\ell_{va_f}$ | Variance of full-meander intrinsic wavelength |
| | $\ell_{sk_h}$ | Skewness of half-meander intrinsic wavelength |
| | $\ell_{sk_f}$ | Skewness of full-meander intrinsic wavelength |
| | $\ell_{kr_h}$ | Kurtosis of half-meander intrinsic wavelength |
| | $\ell_{kr_f}$ | Kurtosis of full-meander intrinsic wavelength |
| Asymmetry | $\mathscr{A}_{av_h}$ | Mean half-meander asymmetry |
| | $\mathscr{A}_{av_f}$ | Mean full-meander asymmetry |
| | $\mathscr{A}_{va_h}$ | Variance of half-meander asymmetry |
| | $\mathscr{A}_{va_f}$ | Variance of full-meander asymmetry |
| | $\mathscr{A}_{sk_h}$ | Skewness of half-meander asymmetry |
| | $\mathscr{A}_{sk_f}$ | Skewness of full-meander asymmetry |
| | $\mathscr{A}_{kr_h}$ | Kurtosis of half-meander asymmetry |
| | $\mathscr{A}_{kr_f}$ | Kurtosis of full-meander asymmetry |
| Curvature | $C_{av}$ | Mean channel curvature |
| | $C_{va}$ | Variance of channel curvature |
| | $C_{sk}$ | Skewness of channel curvature |
| | $C_{kr}$ | Kurtosis of channel curvature |

## Meandering river dynamics

To seek an explanation for the observed disparity in meandering river morphometrics as a function of riparian vegetation density, we also investigated differences in the dynamic evolution of meander planforms over time. Meander evolution is known to be controlled by flow-imparted sediment erosion and deposition at the outer and inner banks, respectively, driving the lateral migration and planform development of meandering rivers over time[1,2].

Quantification of meander planform evolution through DTW (Methods) suggests statistically significant differences in rates of width-adjusted lateral migration ($M_R = M_R^*/W^*$ [yr⁻¹]) between vegetated rivers (used as the control group) and both semi-vegetated and unvegetated rivers ($p$-value < 0.01 for WRS tests performed at 5% significance level). Migration rates relative to channel width in unvegetated rivers ($M_{R_{unveg}} = 0.019 \pm 0.022$ yr⁻¹, median ± standard deviation values reported) are almost twice as high as those in vegetated rivers ($M_{R_{veg}} = 0.010 \pm 0.058$ yr⁻¹), and about 58% higher than in semi-vegetated rivers ($M_{R_{semiveg}} = 0.012 \pm 0.051$ yr⁻¹) (Fig. 4a). This finding corroborates empirical observations that vegetation helps to stabilize river banks against migration[13,23,42] and suggests that reach-averaged time-lapse analyses, as employed herein,

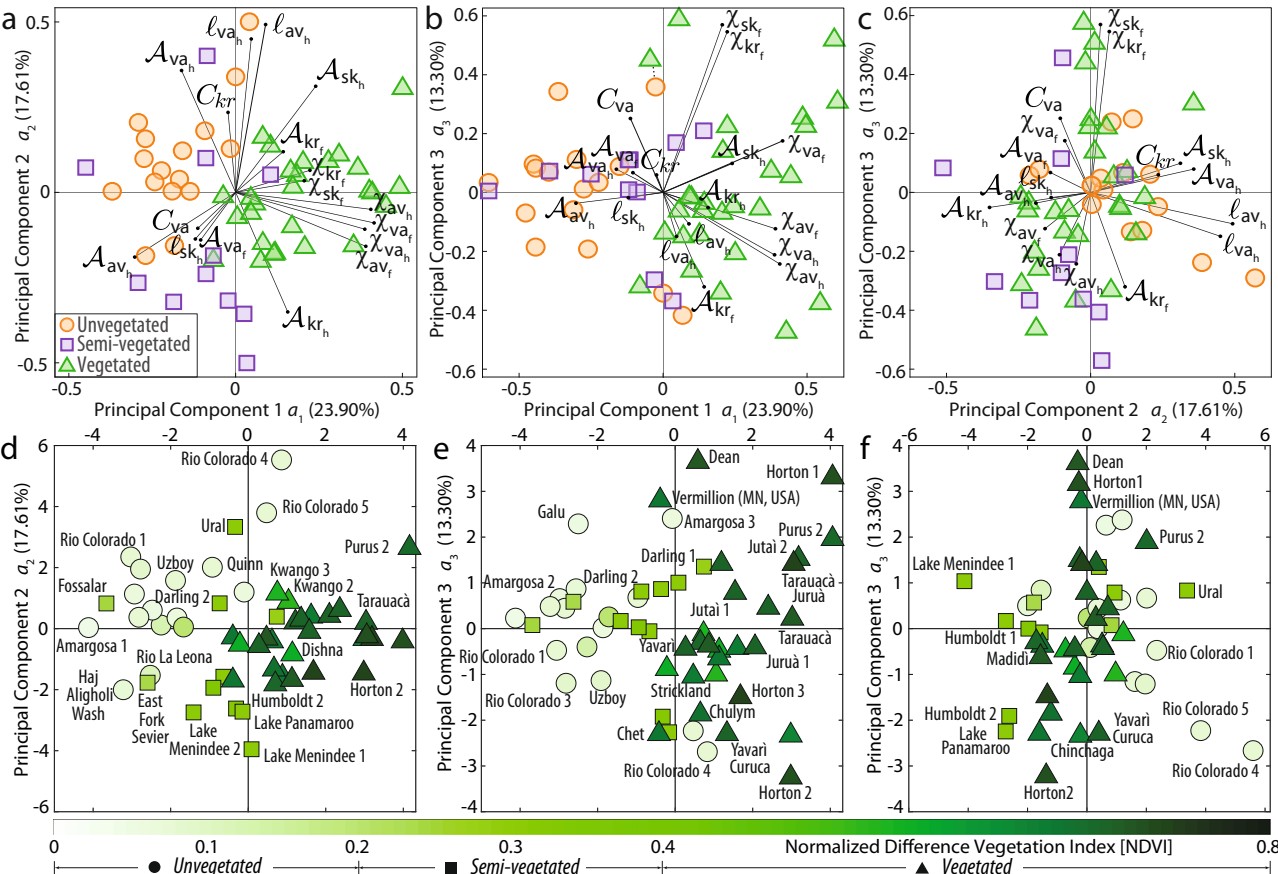

**Fig. 3 | Principal Component Analysis (PCA) of meandering river planform morphometrics. a–c** Biplot of PC loadings and scores for the first three PCs ($a_1$, $a_2$, and $a_3$). PC loadings correspond to the correlation coefficients defining PCs, while scores are the projections of the original data into the PC space. In order to fit in the loading space, PC scores are divided by the maximum absolute value of all scores and multiplied by the length of the loading vectors. The percent of variance explained by each PC is reported along the corresponding axis. **d–f** Score plots resulting from PCA. Colors represent the Normalized Difference Vegetation Index (NDVI), whereas dots, squares, and triangles denote unvegetated, semi-vegetated, and vegetated rivers, respectively. The percent of variance explained by each PC is reported along the corresponding axis.

attenuate differences in characteristic migration rates between unvegetated and vegetated systems when compared to measurements carried out at bend apexes[32,43–45]. More importantly, DTW analysis of river centerlines also allows for examining the functional relationship between channel curvature and river migration. The extensive literature on this topic[1,4,38,46–50]—which focused almost exclusively on relatively large, sand-to-gravel-bedded rivers in vegetated landscapes – has shown that the distribution of $C$ vs. $M_R$ data is significantly scattered, such that a single value of $M_R$ cannot be generally associated with a single value of $C$ (refs. [51,52]; see Supplementary Method 2). Nevertheless, a quasi-linear relationship emerges between $C$ and both the mean and upper values of the $M_R$ distributions, pointing to a strong first-order control of $C$ on meandering river morphodynamics[37,38,44]. This proportionality typically only holds for mildly curved channel reaches (i.e., for width-adjusted curvature values smaller than 0.25–0.5, taken here as $C < 0.3$ for convenience) and breaks down at higher curvatures ($C > 0.3$) where $M_R$ saturates due to the growth of hydrodynamic nonlinearities that effectively limit bank erosion, such as saturation of centrifugally driven secondary flows, enhanced secondary outer bank cells, and flow separation at the outer bank[1,38,46,53,54] (Supplementary Method 2).

In order to explore the effect of vegetation on curvature-driven meander dynamics, we derived the relationship between dimensionless curvature ($C$) and width-adjusted migration rate ($M_R$) for each river in our dataset, further adjusting migration rates by the reach-average spatial lag ($\Delta^*_{CM}$ [m]) between local maxima in channel

curvature and migration rate[37,44,51] (Supplementary Method 2). In all the river classes, a positive linear relation of the type $M_R = \alpha + \beta \cdot C$ is observed for curvature values $C \leq 0.3$ (Fig. 4b), a pattern consistent with theoretical and numerical predictions[37,55–58]. Linear correlations hold both for average and upper values of the $M_R$ distributions, represented here by the 50th ($M_{R_{50}}$) and 95th ($M_{R_{95}}$) percentiles, respectively, of equally sized sets of $M_R$ data binned according to increasing $C$ values[38] (Fig. 4b). Nevertheless, our results show that the coefficient of proportionality ($\beta$), computed for both $M_{R_{50}}$ and $M_{R_{95}}$ data corresponding to $C \leq 0.3$, are consistently larger in vegetated rivers than in unvegetated and semi-vegetated ones (Fig. 4b). Beyond the $C = 0.3$ threshold, the correlations become negative as $M_R$ decreases with increasing curvature, with a steeper decline observed for vegetated rivers compared to unvegetated and semi-vegetated ones. Notably, the latter class of rivers exhibits significant data scattering, to the extent where all the linear relationships are not statistically significant.

These results still generally hold when we break the rivers out of ensemble vegetation classes and treat them individually by considering characteristic NDVI (Supplementary Fig. 4), even without adjusting curvature and migration values to account for the whole set of river widths in our dataset (Supplementary Fig. 5). Minor deviations of some rivers from the general trend described above are justified by variability in sediment grain size[59], bank erodibility[46], and floodplain heterogeneity[39,60], which are not accounted for in our remote-sensing analyses. Moreover, morphodynamic feedbacks that

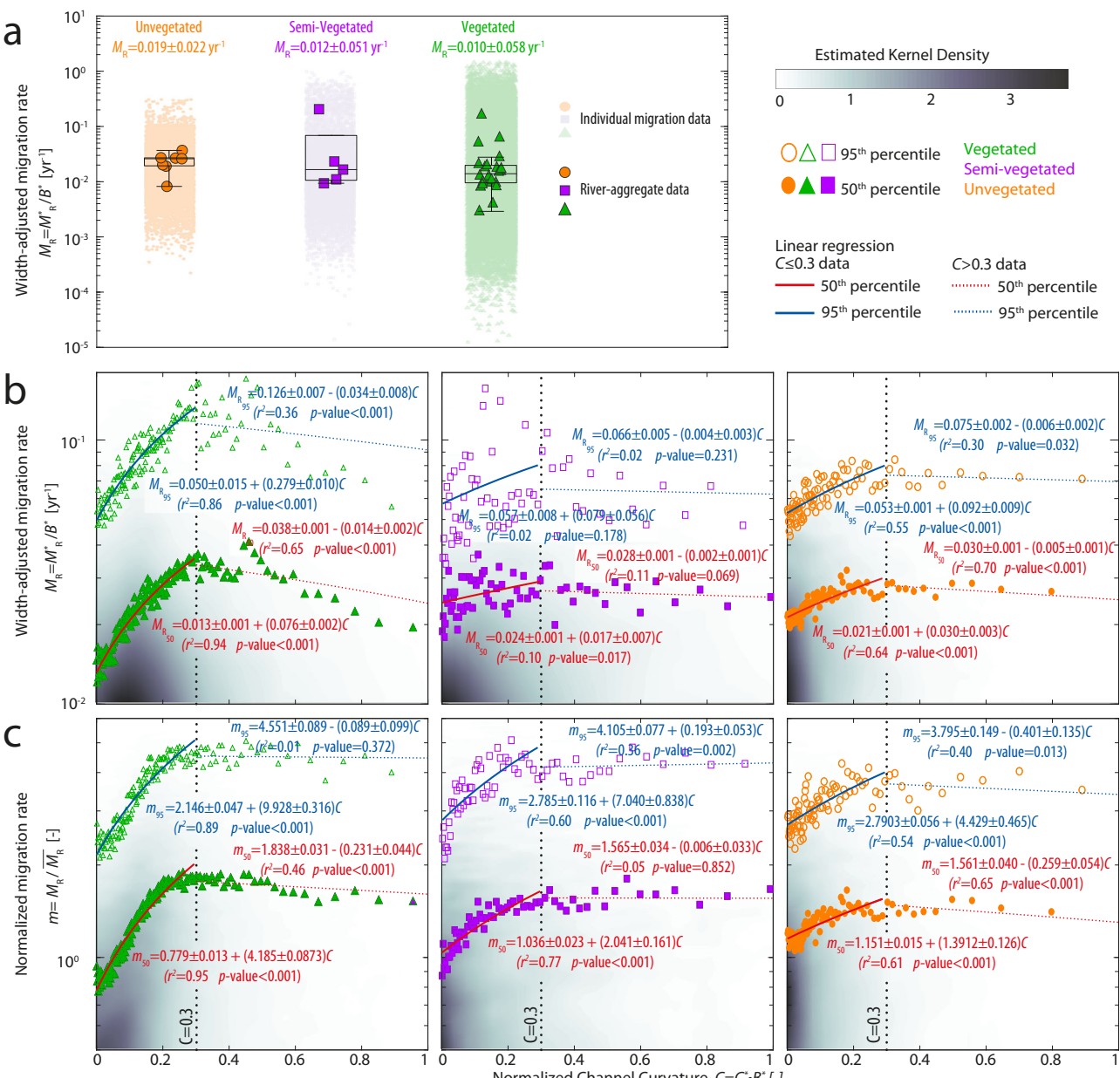

**Fig. 4 | Meandering-river planform dynamics. a** Boxplot of width-adjusted migration rates $M_R = M_R^*/B^*$ ([yr$^{-1}$]) for unvegetated, semi-vegetated, and vegetated rivers. Reduced-size shaded markers in the background represent data of individual meander bends, whereas larger markers denote aggregated data for each river in our dataset. The median is denoted by the central mark, while the lower and upper boundaries of the box represent the 25th and 75th percentiles, respectively. The whiskers reach out to encompass the most extreme data points that are not identified as outliers. Ensemble median ± standard deviation values are also reported. **b** Width-adjusted migration rates $M_R$ plotted against width-adjusted channel curvature $C = C^* \cdot B^*$ for vegetated, semi-vegetated, and unvegetated rivers in the ensemble. **c** Normalized width-adjusted migration rates $m = M_R/\overline{M_R}$ ([-]), where $\overline{M_R}$ denotes the reach-average migration rate of any given river in our dataset, are plotted against $C$ for vegetated, semi-vegetated, and unvegetated rivers in the ensemble. Data reported in **b**, **c** are corrected for the reach-averaged value of the spatial lag between maximum migration and curvature ($\Delta_{CM}^*$ [m]). Empty and filled markers represent the 95th and 50th percentiles of both the $M_R$ and $m$ distributions, obtained by binning together sets of $n$ data points equal to 1000 for vegetated rivers and 250 for both semi-vegetated and unvegetated rivers, respectively. Regression lines are reported separately for the 50th (in red) and 95th (in blue) percentiles of migration distribution, as well as for migration corresponding to $C$ data smaller (continuous lines) and larger (dotted lines) than the critical $C = 0.3$ threshold. Gray-shaded areas in the background represent the bidimensional kernel density estimates of data points.

arise from vegetation patterns and habits likely contribute to variability in ($M_R$, $C$) that emerge even among rivers with similar NDVI values (Supplementary Figs. 4 and 5), and add to the variability observed in semi-vegetated river data (Fig. 4b). Plant density, as well as channel width and depth relative to plant stem diameter and rooting depth[13,15,61–63], have been shown to influence channel dynamics. In sparsely vegetated landscapes, shrubby arborescent, drought-resilient plants will encroach into the moister and nutrient-

rich thalweg zone of ephemeral channels, where groundwater is more accessible[15,21]. Especially in rivers narrower than 10 m, such clusters of in-channel vegetation can enhance local scour and bank erosion, disrupt curvature-induced helical flow that sustains lateral migration, and limit meander growth by facilitating braiding and chute cutoffs[15,62] (formed when the river cuts a new bypass channel through its own point bar[64]; see Supplementary Fig. 6). In contrast, deep-rooted trees can sustain large single-thread meandering

channels (width $> 10^2$ m)[15,63] in densely forested floodplains, aiding flow confinement through enhanced bank-erosion resistance and facilitating the continued development of highly sinuous meander bends until neck cutoff. On average, larger rivers are associated with higher NDVI values (Supplementary Fig. 7), but our analysis of normalized morphometric characteristics shows that the effects of vegetation regime on meandering dynamics are independent of channel size (Fig. 2a–d).

### Curvature-driven meander dynamics enhanced by riparian vegetation

To investigate variations in channel mobility while avoiding highly mobile rivers to systematically bias our data, we further normalized $M_R$ by dividing it by the average migration rate of the corresponding river reach ($\overline{M_R}$) (Fig. 4c). Importantly, this procedure also allows for filtering out spurious correlations due to inherent hydrological and sedimentological variability (e.g., flow intermittency and changes in bank erodibility) broadly related to site-specific climatic and environmental conditions.

For $C \leq 0.3$, the normalized width-adjusted migration rates $m = M_R / \overline{M_R}$ ([-]) increase linearly with curvature regardless of vegetation density and whether the 50th and 95th percentiles of $m$ data are considered (Fig. 4c). Normalizing migration rates significantly reduces data scatter, especially for semi-vegetated rivers, and makes all linear relationships strong and statistically significant ($p$-value < 0.001). Most importantly, the slope ($\beta$) of linear regression lines increases progressively from unvegetated to semi-vegetated and vegetated rivers, both when considering the median ($\beta_{50_{vege}} = 4.19 \pm 0.09$; $\beta_{50_{semi}} = 2.04 \pm 0.16$; $\beta_{50_{unv}} = 1.39 \pm 0.13$) and 95th percentile of $m$ data ($\beta_{95_{vege}} = 9.93 \pm 0.32$; $\beta_{95_{semi}} = 7.04 \pm 0.84$; $\beta_{95_{unv}} = 4.43 \pm 0.47$). In contrast, the regression-line intercept ($\alpha$) is larger in semi-vegetated and unvegetated rivers, suggesting relatively high rates of meander migration even at low curvatures ($\alpha_{50_{vege}} = 0.78 \pm 0.01$; $\alpha_{50_{semi}} = 1.04 \pm 0.02$; $\alpha_{50_{unv}} = 1.15 \pm 0.02$; $\alpha_{95_{vege}} = 2.15 \pm 0.05$; $\alpha_{95_{semi}} = 2.79 \pm 0.12$; $\alpha_{95_{unv}} = 2.79 \pm 0.06$). For $C > 0.3$, a consistent plateauing in $m$ values is observed for all three vegetation classes. The regression lines become nearly horizontal, indicating lack of correlation between migration rates and curvature, despite some regressions still being statistically significant (Fig. 4c). Whereas saturation of migration rates for $C > 0.3$ appears unrelated to vegetation and is likely better explained by the complex flow structures that arise in sharp bends[46,53,65–67], the observed differences in $m$ for $C \leq 0.3$ can be attributed to the lack (or sparsity) of riparian vegetation. In particular, the diminished proportionality between migration rate and curvature, coupled with the relatively higher migration rates observed in semi-vegetated and non-vegetated rivers (as curvature approaches zero) underscore the impact of lack of vegetation on bank stability and sediment transport processes. The lack of vegetation enhances bank erosion through slump block collapse, which can occur even at low curvatures[15]. Additionally, it fosters hydrological connectivity between the river and its floodplain by limiting the formation of high-relief levees[29]. The reduced topographic prominence of levees facilitates sediment exchanges between channel and floodplain and the formation of erosional rills along the banks during waning flood stages, ultimately resulting in lateral channel infilling driven by return flows[29]. Consequently, pronounced sediment deposition is observed even at the outer (concave) banks of meanders, where larger water depths and sustained erosion should be expected if riverbanks were vegetated[15,29]. In addition, local perturbations of river morphology induced by widespread chute cutoffs in barren and poorly vegetated settings likely lead to accelerating migration and channel widening both locally and nonlocally[68,69], thus further enhancing $m$ and data scatter at low curvature values that are typical of chute cutoffs. All of these processes are likely to reduce correlation between channel curvature and bank erosion (and

related river migration) as the density of riparian vegetation decreases, thus explaining our observations.

## Discussion

The persistent patterns and relationships emerging from our analyses collectively demonstrate that vegetation critically affects meandering river morphodynamics by subtly but systematically modifying the relationship between river lateral migration and curvature. Perhaps surprisingly, this effect is most evident at relatively low curvatures ($C \leq 0.3$), where semi-vegetated and unvegetated rivers reflect relatively high rates of meander migration (Fig. 4b, c; Supplementary Figs. 4 and 5). Where curvature is large ($C > 0.3$), in contrast, any vegetation-induced effect can hardly be discerned because migration rates and dynamics are similarly independent of channel curvature regardless of riparian vegetation density.

We emphasize that values of $C \leq 0.3$ are prevalent along meandering river courses[1,33,37] (Fig. 2k), such that vegetation is likely to leave a mark on meandering river morphodynamics. By stabilizing the low-curvature reaches of meandering rivers, vegetation effectively pins down meander inflections, allowing meander bends to gain both curvature and sinuosity as the channel migrates laterally. Such interpretation is supported also by our insights on meander morphometrics suggesting that denser riparian vegetation correlates with the continued growth of meanders until late-growth stages[40,41], when bends become highly skewed and sinuous and are more likely to shortcut themselves through neck cutoff[40]. Meander neck cutoffs are comparably less common in non-vegetated and semi-vegetated rivers (Fig. 1c and Supplementary Fig. 6), which in contrast feature numerous chute cutoffs that prematurely interrupt meander growth and sinuosity development (Supplementary Fig. 6). Since barren and poorly vegetated river floodplains are conducive to chute formation owing to the reduced erosion resistance (with sparse and drought-resistant plants further acting as erosion nuclei for the development of chute channels)[28,70,71], a causal correlation between the lack of vegetation and reduced meander morphodynamic maturity might be implied. However, the morphology of meandering rivers in our dataset cannot be unambiguously differentiated in terms of vegetation density alone. Morphological differences captured by PCA become more subtle when the underlying NDVI values are considered rather than the three binned classes for vegetation (Fig. 3d–f), despite the transition in NDVI values from unvegetated to vegetated rivers being relatively smooth. This phenomenon may be attributed to environmental factors beyond lack of vegetation, which might render unvegetated rivers morphologically different from vegetated ones. Notably, some of these environmental factors are known to co-vary with the abundance of vegetation, thereby potentially confounding any morphological separation based on vegetation density alone. For instance, the Aridity Index (AI) (i.e., the ratio between precipitation and evapotranspiration; see Methods), strongly correlates with NDVI (Supplementary Fig. 8), with unvegetated rivers being also characterized by lower AI (i.e., by higher aridity). Yet, while aridity could influence morphodynamics by modulating river flows and formative discharge conditions[72–74], the AI alone cannot fully account for the entire variance in the dataset. As a matter of example, the semi-vegetated Fossalar River[75] (Iceland) clusters with low-AI river data in the negative $a_1$ half-space, despite having the highest AI of all investigated systems and being a perennial river, with baseflow throughout the year (Supplementary Fig. 9). Indeed, besides vegetation and aridity, hydrological regime and sediment flux have both been cited as exerting controls on meander planforms[43,44,72–78]. Whereas unvegetated rivers are typically found in arid or semi-arid climate zones[15,21], many vegetated rivers flow through humid tropical or temperate continental settings. This aspect translates into markedly different hydrological and sediment transport regimes[43,78–80]. Flashier hydrological and

sediment-transport regimes in arid and semi-arid settings may also further facilitate meander chute cutoffs, compounding the effect of reduced riparian vegetation density. Besides, meandering streams in modern barren environments are found almost exclusively along low-gradient (slope ~$10^{-5}$–$10^{-4}$) terminal fluvial fans where relatively high rates of vertical aggradation increase the frequency of river avulsions[5,15,25,81]. Frequent avulsions prematurely disrupt meander evolution, thereby limiting the development of sinuosity and curvature in as much as meander chute cutoffs do. Hence, whereas our findings regarding the effects of vegetation on meander morphology dovetail with many complementary explanations for varying meander morphometrics, more frequent avulsions in modern unvegetated and sparsely vegetated settings are a confounding factor that makes it challenging to infer direct causation between vegetation and meandering river planforms. This notion however does not diminish the role that vegetation plays in affecting meandering river dynamics. Notably, the differences we observed in the functional relationship between river lateral migration and curvature remain unaffected by the disparity in avulsion regimes among vegetated and unvegetated rivers.

All in all, vegetation stabilizes river banks, slows lateral migration rates, and enhances the control of channel curvature on bank migration at low-to-moderate curvatures. Without such a stabilizing effect, in contrast, unvegetated and poorly vegetated meandering rivers wander more even at low curvatures. Importantly, the proper normalisation of migration rates we applied in our analyses ensures that these dynamics are insensitive to aridity and other allied environmental variables that broadly correlate with vegetation density (e.g., sediment fluxes). Hence, at least some of the observed correlation between morphometrics and vegetation (Figs. 2 and 3) could be causal, although the critical disparities in avulsion frequency between modern vegetated and unvegetated rivers prevent drawing unequivocal conclusions regarding the influence of vegetation on meandering river planforms.

The vegetation-related changes in curvature-driven planform dynamics that we observe are likely to be preserved in the stratigraphic record, necessitating their inclusion in depositional models for unvegetated single-channel rivers[19,22,29] to enhance the identification of ancient unvegetated meandering rivers in the rock record and to improve paleohydrological reconstructions on early Earth and Mars. Furthermore, our results provide new insights into channel mobility and floodplain reworking in meandering rivers—a key aspect to understanding watershed-scale biogeochemistry, particularly in relation to weathering processes and floodplain carbon stocks and fluxes[5–8].

## Methods

The dataset analyzed in this study includes 54 meandering rivers encompassing different climate and geological regions, from polar to hyperarid, thereby ensuring representativeness for distinct hydrological regimes, sediment grain sizes, and, importantly, densities of riparian vegetation. The selected rivers also ensure an inclusive sampling strategy concerning geographical distribution (spanning most continents; Fig. 1a) and river size (covering three orders of magnitudes in width; Fig. 2a–d).

### Extraction and analysis of meandering river planforms

To obtain planform morphometrics of individual meandering reaches, we first hand-digitized riverbank lines in the QGIS environment based on freely available georeferenced aerial and satellite images. The latter include several different products available through Google Earth Pro, SASPlanet, the U.S. Geological Survey Earth Explorer portal, and the QuickMapService QGIS plugin. Additionally, we included in our dataset the data collection provided by Sylvester et al.[37] for vegetated meandering rivers in the Amazon basin (Fig. 1a).

In order to ensure data homogeneity among distinct fluvial settings, we only considered single-thread meandering river reaches that (i) contain at least 35 consecutive bends, such that a sufficiently long train of meanders can be analyzed; (ii) are found sufficiently far away from the coastline in order not to be affected by backwater and/or tidal effects; (iii) have not been significantly modified or engineered by humans.

Digitization of rivers in poorly vegetated settings includes also some abandoned (i.e., non-active) reaches for which planform morphometrics could be identified from aerial images. In the case of the Rio Colorado (Salar de Uyuni, Bolivia) and the Amargosa River (California, USA) (Fig. 1a), the analyzed reaches, though inactive, are still recent enough such that they cannot represent different riparian states, e.g., from past interglacials[25,82]. The dataset also includes the inactive Uzboy River in the Karakum Desert (Turkmenistan), which formed between the Upper Pliocene and the Preglacial Quaternary under an arid palaeoclimate[21], although estimates of vegetation content during its active phase are still debated[17,83].

To ensure a baseline of accuracy for all morphometric data collection, riverbanks were digitized using aerial scenes with a minimum ground resolution corresponding to approximately 5% of the average river width (e.g., images with 30 m ground resolution were used only for digitizing river larger than 600 m, whereas images with resolutions equal to or higher than 0.5 m were employed for 10-meter-wide rivers). For vegetated rivers, the bank line position was determined based on vegetation boundaries. In unvegetated and sparsely-vegetated rivers, the position of outer (concave) banks was typically clear as the cut banks are sharp and form an abrupt change of angle with the adjacent floodplain, whereas bank position along the gently-sloping meander inner (convex) side was identified by the accumulation of debris at the point-bar top, forming a line parallel to the channel path[5,21] (Supplementary Fig. 6).

Freely available, high-resolution topographic data for the unvegetated Amargosa River (Death Valley, California, USA) and the semi-vegetated meandering systems developed in the Menindee and Panamaroo lakes (Darling River basin, New South Wales, Australia) were employed to test the accuracy of width measurements derived from aerial imageries ($B^*_{sat}$ [m]) against bankfull width computed from topographic data ($B^*_{topo}$ [m]). Results show a strong fit between the two width datasets ($B^*_{topo} = 1.01 \cdot B^*_{sat}$; $r^2 = 0.98$, $p$-value < 0.001) measured at constant increments equal to 10 channel widths along the studied channel reaches (Supplementary Fig. 10).

Once riverbanks were digitized, we derived channel centerlines through a standard skeletonization procedure[33]. Channel centerlines were smoothed by means of a Savitzki-Golay lowpass filter to avoid numerical discontinuities and then resampled using standard cubic spline-fit polylines with a spatial resolution approximately corresponding to one-tenth of the channel average width[37]. To isolate individual meander bends we used a semi-automated procedure based on the computation of local channel-axis curvature $C^*$ ([$m^{-1}$]) (Supplementary Fig. 11). Specifically, for any given centerline point $\{x^*(s^*), y^*(s^*)\}$, we computed $C^* = d\theta(s^*)/ds^*$, where $s^*$ ([m]) denotes the channel centerline curvilinear coordinate, $\{x^*(s^*), y^*(s^*)\}$ represent the coordinates of an arbitrary axis point in a Cartesian reference system, and $\theta(s^*)$ is the angle formed by the tangent to the channel axis and an arbitrarily fixed direction[33,37]. After computing $C^*$, a Savitzky–Golay low-pass filter with a fixed polynomial order of 3 and a frame length of 21 centerline points was applied to further smooth noise in the original $C^*$ signal. Subsequently, half (full) meander bends were identified as the portion of the channel between two (three) consecutive inflection points (i.e., points where $C^* = 0$). Meander apexes were also identified as points corresponding to local curvature maxima between inflections (Fig. 1c and Supplementary Fig. 11). We note that the low-pass filtering of $C^*$ signal resulted in the automatic deletion of some "spurious" inflection points delimiting meander bends characterized by

limited length and sinuosity, which are typical in double-headed meander bends featuring multiple local maxima in the $C^*$ signal (Supplementary Fig. 11).

Morphological features of individual meander bends were finally characterized based on several morphometric parameters (Fig. 2, Inset), namely: meander intrinsic wavelength ($\ell^*$ [m]), which is the along-channel distance between inflection points; meander cartesian wavelength ($L^*$ [m]), defined as the planar distance between meander inflections; meander sinuosity χ = $\ell^*/L^*$ ([-]); meander amplitude ($A^*$ [m]), measured as the maximum point-line distance between any centerline point and the line connecting the two meander flexes; meander radius ($R^*$ [m]), defined as the radius of the best-fitting circle obtained by considering all meander centerline points[50]; and meander asymmetry index, computed as $\mathscr{A} = (\ell^*_u - \ell^*_d)/\ell^*$ ([-]) where $\ell^*_u$ and $\ell^*_d$ denote the distance between the meander apex and its upstream and downstream endpoints (Fig. 2, Inset), respectively, so that negative (positive) values of $\mathscr{A}$ indicate meanders skewed in the upstream (downstream) direction.

To directly compare rivers of different sizes, all the dimensional morphometric variables (i.e., those denoted with superscript asterisks) were normalized with the average channel width $B^*$([m]) measured along individual meander bends (e.g., $L = L^*/B^*$; $\ell = \ell^*/B^*$; $C = C^*\cdot B^*$; $A = A^*/B^*$; $R = R^*/B^*$).

## Riparian vegetation density and aridity

To characterize riparian vegetation density, we utilized multi-annual statistics of the remotely-sensed Normalized Difference Vegetation Index (NDVI) computed through Landsat 7 Collection 1 Tier 1 8-Day NDVI composites with a ground-spatial resolution of 30 m, accessed through Google Earth Engine. The composites are derived from Tier 1 orthorectified scenes, using the computed top-of-atmosphere (TOA) reflectance, and are generated from all the scenes in each 8-day period beginning from the first day of the year and continuing to the 360th day of the year. As such, the final composite for a given year, starting from day 361, has a 3-day overlap with the initial composite of the subsequent year. Each composite includes all images from its respective 8-day period, with the most recent pixel serving as the composite value.

The NDVI is computed based on the Near-InfraRed (NIR) and Red bands of each scene as NDVI = (NIR - Red)/(NIR + Red) and ranges in value from −1.0 to +1.0. Snow filtering was also applied for rivers in continental and arctic climate zones based on a fixed normalized difference snow index (NDSI) threshold (NDSI = 0.4), where NDSI represents the normalized difference between green spectral bands and the shortwave infrared.

For our analyses, we created a percentile composite of image collection wherein each pixel contains the 90th percentile of NDVI (NDVI90) values computed over a 20-year timespan (i.e., from 1999 to 2019) (Supplementary Fig. 12). Then, for each river in our dataset, we estimated the characteristic floodplain NDVI as the median of NDVI90 computed within an area obtained by buffering the river centerline with a buffer distance equal to 10 times the average river width (Supplementary Fig. 12). Finally, we categorized our rivers into three classes based on NDVI values: unvegetated (NDVI ≤ 0.2), semi-vegetated (0.2 < NDVI ≤ 0.4), and vegetated (NDVI > 0.4). For the meandering streams found in Lake Menindee and Lake Panamaroo (New South Wales, Australia), which were found to be flooded for most of the observation period, we assumed the same NDVI value calculated for the floodplain of the nearby Darling River, from which the lakes originate (Supplementary Fig. 13).

We used an approach akin to the method used for calculating NDVI to calculate the Aridity Index for the examined river reaches. The median Aridity Index (AI) was computed, within the same buffer area employed to derive NDVI statistics, based on the freely available "Global Aridity Index and Potential Evapo-Transpiration (ET0) Database v3" provided as high-resolution (30 arc-seconds) global raster data for the 1970–2000 period[84].

## Channel migration rate

Meander lateral migration was analyzed for a subset of rivers for which active migration was detected based on pairs of images available at adequate resolution. Lateral migration was measured at fixed increments $\Delta x^*$ no larger than the average river width along the channel centerline using a dynamic time-warping algorithm (DTW) implemented in R (ref. 85) and performed through the QGIS software (v.3.6.3) processing tool (Supplementary Fig. 11). Originally developed to correlate time series, DTW represents a state-of-the-art method to compute channel lateral migration in dynamic single-thread meandering systems thanks to improved estimates of bank migration trajectories compared to typical proximity algorithms such as nearest neighbor and inverse-distance weighted[37,86]. The DTW algorithm employs a cost matrix to minimize the sum of distances, rather than individual distances, between two consecutive river centerlines. Specifically, DTW alignment of two consecutive river centerlines is performed through a Euclidean distance matrix for corresponding centerline points, augmented with the third dimension of curvature at those same points. That is, for each pair of $i$ and $j$ point along the original and final river centerline, the matrix of following distance is computed:

$$d^*_{i,j} = \sqrt{\left(x^*_i - x^*_j\right)^2 + \left(y^*_i - y^*_j\right)^2 + \lambda^2\left(C^*_i - C^*_j\right)^2} \tag{1}$$

where $\lambda$ is a multiplier weighting parameter for curvature values. In this way, the sum of distance is weighted by both the spatial distance and the similarity in local channel curvature, so that the algorithm can effectively monitor changes in both channel centerline position and curvature[26,37,38,44] (Supplementary Fig. 11). While this approach is similar to a simple nearest neighbor search in principle, its results differ from the latter as DTW effectively avoids large gaps between correlated centerline points (see example in Supplementary Fig. 11).

After DTW computations, we identified and masked meander bend cutoffs within each river reach to exclude them from subsequent analyses (Fig. 1d). Then, following the same procedure used for filtering $C^*$, migration ($M^*$ [m]) data were smoothed using a low-pass Savitzky-Golay filter to reduce data noise (Supplementary Fig. 11), and migration rates were computed as $M^*_R = M^*/\Delta t$ ([m yr$^{-1}$]), where $\Delta t$ denotes the time difference, in years, between the dates when river centerlines were acquired. Finally, in order to account for the spatial lag between curvature and migration maxima (see Supplementary Method 2 for a thorough discussion), we shifted the $M^*$ signal upstream by a length corresponding to the reach-averaged curvature-migration lag $\Delta^*_{CM}$ (Supplementary Method 2). In this way, the functional correlation between channel curvature and lateral migration can be studied in a physically sound fashion.

## Data availability

The source data generated in this study have been deposited in a public database (ref. 87) [https://doi.org/10.5281/zenodo.10393446]. Previously published, freely available data were also used for this work (ref. 37).

## Code availability

The code used for computing meander migration through Dynamic Time Warping is freely available from a GitHub repository (ref. 88) [https://doi.org/10.5281/zenodo.10657996] as well as from the source data listed above.

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

## Acknowledgements

This study is funded by the European Union—NextGenerationEU and by the University of Padua under the 2021 STARS Grants@Unipd programme, project "TiDyLLy- Tidal networks dynamics as drivers for ecomorphodynamics of low-lying coastal area" (to A.F.), by a Visiting Scientist Scholarship from the University of Padua (to A.I.), and by the HYDROSEM Project sponsored by the CARIPARO Foundation (to M.G.). AF, MG and ADA are supported by the Italian Ministry of University and Research (MUR) through the project titled "The Geosciences for Sustainable Development" (Budget MUR - Dipartimenti di Eccellenza 2023-2027; Project ID C93C23002690001). A.I. is supported by a Discovery Grant from the Natural Sciences and Engineering Research Council of Canada (RGPIN-2016-5720). The authors thank Chris Paola for valuable discussions about this work and James Pizzuto for reviewing an earlier manuscript version.

## Author contributions

A.F. and A.I. designed research and developed analytical tools. A.F. and S.F. digitized riverbanks and analyzed meander planform data. A.F., A.I., S.F. and D.T. analyzed migration data. A.F. and D.T. prepared figures. A.F., A.I., E.D.L. and M.G.L. drafted the paper. M.G., L.C. and A.D.A. contributed critically to the drafts. All authors gave final approval for publication.

## Competing interests

The authors declare no competing interests.
