## [Peer Review File · Nature Communications]

Vegetation enhances curvature-driven dynamics in meandering riversREVIEWER COMMENTS

Reviewer #1 (Remarks to the Author):

Specific comments:

Line 24: The connotation of “morphodynamic maturity” will likely not have meaning to the reader until it is defined.

Line 28-29: It is unclear what is implied by “enhances the proportionality between migration rate and channel curvature”, i.e., a higher correlation coefficient, a higher regression slope, or a higher intercept? It is unlikely to be the latter since the migration rate is lower according to the first clause of the sentence.

Line 29: The antecedent “effect” is unclear for this sentence. Is it the reduced “rate of river lateral migration”? Also, what is “obscured”? The rate of migration?

Line 31: Eliminate “Hence,” because this has not been demonstrated in the abstract. The sentence works without this.

Lines 44-46: The text leaves unresolved the question of why heterolithic strata are uncommon prior to the development of vascular plants, given that vegetation is asserted to be not necessary for meandering behavior. Is this issue addressed later in the paper?

Line 62: It is unclear what would be implied by the inverse relationship implied by “vice-versa”. Would meandering somehow lead to development of vegetation?

Line 78 and lines 418-450: The present authors are probably not responsible for the misnomer “dynamic” in dynamic time warping. Based on the description, this methodology is at best “kinematic” because it is not based on mechanistic, deterministic forward modeling.

Line 88-89: Because “x” is a scaled variable in the equation, it probably should be designated as “x*”.

Line 91-92: “fairly constant” is not quantifiably interpretable. Does it imply statistical equality of values?

Line 96: Clarify what is meant by [-] .

Line 102-103: “significant differences” should be “significant differences in morphometric relationships”

Line 132: The criteria for this downsizing is not specified. This is not a crucial issue, but could be mentioned in the supplementary materials or methods.

Line 146: Here again, “maturity” of bends is mentioned without precise definition.

Line 170: Is “impulsive” a well-recognized descriptor of sediment transport behavior?

Lines 183-186: I’m not sure why the effect of vegetation on chute cutoffs is a “confounding” factor and makes it challenging to infer causation. Cutoffs are probably the dominant factor in explaining the lower average sinuosity in arid-zone channels due to sparse floodplain vegetation.

Line 211: something is missing in the phrase “pointing to a strong first-order control of C meandering river morphodynamics”

Line 217 and Supplementary Material 2: It is unclear what is meant by “correcting migration rates by the average spatial lag”. The effect of the lag is clear in Supplementary Figure 10. Is the lag used to “correct” the data the average of the individual lags applied to the reach as a whole, or is the lag determined and applied to each bend individually? Parenthetically, the term “corrected” should probably be “normalized” or “adjusted”. In Supplementary Figure 13 all the lags are positive, presumably implying a sub-resonant regime. Was this true for all rivers studied?

Materials and Methods: Extraction and analysis of meandering river planforms: As is acknowledged in

the discussion care needs to be taken to remove “noise” related to the digitizing procedure in determining the generalized planform. The discussion is generally clear about the procedures, but some detail is lacking due to reliance on previous studies. One of the most critical steps in segmenting natural channels is defining the inflection points. The issue is illustrated in Supplementary Figure 10, where near the north arrow there is a slight inward inflection of the adjacent long bend. If the plotted centerline between iv and v is correctly plotted there should be two intermediate inflection points and one half-bend that are not evident in part B of this figure. This issue only becomes important if including those inflection points would significantly affect the subsequent analyses. Have the authors, or previous studies, evaluated the influence of planform generalization on resulting morphometric parameters?

General comments:

The role of sediment supply characteristics in meandering behavior is not discussed in this paper. In this comment I rely on informal observations that may not be universally applicable. Meandering in arid environments seems to occur only when the supplied sediment contains a high fraction of mud relative to sand and gravel. Sand or gravel-dominated channels tend to be braided (this also applies to glacially-fed channels). The “point bars” of these mud-dominated channels are also cohesive. The presence of vegetation in more humid conditions allows stable meandering in sand and gravel-dominated settings which would be braided in more arid environments. Thus the characteristics of sediment supply may be an uncontrolled factor in the present analysis that is indirectly associated with vegetation.

The high frequency of chute cutoffs in channels with low density of vegetation may influence average migration rates due to the perturbation of the flow field, which generally results in rapid adjustments of the planform and locally high migration rates. This is very evident in simulations of meandering where neck cutoffs result in fast migration, sometimes with reverse direction of migration. This is largely compensated for in the present study by considering in detail only the low-curvature bends, but could still have some influence on low-curvature bends.

Overall evaluation: This is an important and informative evaluation of the role of vegetation on meandering behavior. The methodology and analyses are largely clearly explained. Most of the detailed comments can be easily addressed. The issues raised in the general comments are largely beyond the scope of the paper, but the authors should assess whether they have implications for interpretations in the paper

Reviewer #2 (Remarks to the Author):

This is a very timely work that reveals the role of vegetation in meandering river development. Multiple controls have been previously proposed for the long-standing debate about the necessary conditions for meandering river development. Among these controls, vegetation has stirred up a lot of discussions recently, but how specifically does vegetation affect the morphology and dynamics of meandering river have not been systematically explored in sufficient details until this current manuscript. This work documented a list of key geometric attributes of meandering river planform morphology across a wide range of riparian vegetation densities. Statistical analyses of these attributes reveal that the impact of vegetation on meandering morphology and dynamics is subtle but significant. Moreover, the author

found the range of curvature that distinct vegetation control operates. These results have important implications for the role of riverine environments in global carbon cycle and reconstruction of the morphology and dynamics of ancient rivers on Earth and Mars.

There is one missing aspect of the dataset analyzed that could potentially hinder the application of the results from this study. It seems that the rivers in the analyzed dataset collected by the authors are all small to intermediate-sized rivers, and most large lowland rivers are not included. This raises the question whether the findings of the study apply to large lowland meandering rivers as well. I speculate that the impact of vegetation on large rivers is limited as the flow depth of these rivers extends far below the roots of vegetations. This does not negate the results of this study but suggests that there might be a river size limitation on the role of vegetation on meandering river morphology and dynamics. In order to maximize the application of this work, I would suggest the author to consider the following:

(1) Include some large rivers in the dataset. Whether the large lowland rivers follow the trends demonstrated in this study will be useful information. If they do, then it supports that the impact of vegetation is also important for large rivers. If not, there might be a river-size limit for vegetation to affect the morphology and dynamics of meandering river. This is not to underrate the importance of small- to intermediate sized rivers, as they comprise a large portion of global rivers.

(2) Characterize the size of the rivers in the dataset in terms of slope and discharge and provide the overall range/distribution of these metrics. This will help to put the size of the rivers in the dataset into a spectrum so that in case no large river will be included in the analysis, at least it shows the slope and discharge range that the current results are applicable to.

Some minor points for the author to consider:

Is there a threshold curvature used for identifying cross-over point? For example, there seems to be low curvature bends that are part of a compound bend but not considered as a bend defined by cross-over point in figure 1C (e.g., upper section of Humboldt river below the 7th cross-over point, and mid-section of the Rio Horton river below the 13th cross-over point). It seems that the current method skips some of the low curvature bends.

Specific comments:

Line 71: what about different types of vegetation, which may contribute to the spread of data in the analysis¹.

Line 177 “low-gradient terminal fluvial fans...”

Could you provide the actual slope range for the studied rivers.

Line 212 “reaches ($C < 0.3$)...”

How is this threshold value of curvature determined? Seems the curvature for peak adjusted migration rate for semi-vegetated and unnegated rivers is closer to 0.2 (Fig. 4B, C).

Line 236 “morphodynamic feedbacks that arise...”

What specific feedback this is referring to?

Line 249 “...independent of channel size”

Again, what about large rivers.

Chenliang Wu
Postdoc Researcher
Tulane University

References

1. Zhu, L., Chen, D., Hassan, M. A. & Venditti, J. G. The Influence of Riparian Vegetation on the Sinuosity and Lateral Stability of Meandering Channels. *Geophysical Research Letters* 49, e2021GL096346 (2022).

Reviewers #3 and #4 – Co-Review (Remarks to the Author):

Review of Vegetation Signature on River Meander Morphology and Dynamics by Finotello et al.
submitted to Nature Communications

Finotello et al. provide a comprehensive analysis to show how vegetation influences river meandering. The manuscript is well written and the authors should be applauded for their detailed science and valuable data its created. The work is significant. The authors find that rivers with unvegetated and vegetated banks have different planform sinuosities and bend asymmetries—suggesting that bends don't reach maturity in unvegetated settings. They also use a novel analysis and show that while rivers with all types of bank vegetation have consistent curvature vs. migration rate relationships, unvegetated rivers have faster bank migration rates at low curvature. The methodology and data analyses are very sound, and the results are of broad interest to fluvial geomorphologists and geologists that keep up with the debate on the role of vegetation in river meandering. With that said, there are a few overarching difficulties I had with the present version of manuscript that include: 1) more discussion connecting the morphometric and migration rate analyses, 2) a need to better justify methodological choices, and 3) figure simplification and readability.

There are two related, but separate analyses included in the manuscript. The first half of the article focuses on a morphometric analysis of river planform. They utilize and report on two different statistical approaches (population inference and PCA). The second half of the paper is a novel test of existing empiricism and theory that for the first time in my knowledge, looks at curvature and meander migration for both unvegetated and vegetated rivers. However, there is little structure connecting these two analyses. Some notes on this:

- The “Other Impacts on Planform Morphology” section (L150) is indicative of the difficulty the text had in bridging the two analyses. I found this section out of place as it was a discussion describing drivers of planform morphology that aren't explicitly vegetation without a clear section devoted to discussing the role of vegetation itself on planform morphology.
- Further, the opening of the “Meandering river dynamics” section (L189) is too generic and can use more justification as to why were are looking at morphometrics and dynamics in the same analysis. Instead, the paragraph (L189-L203) spends valuable text comparing to previous literature, which while it is great to see that it lines up, could be used to tie the analyses together.

- The primary connection between the two analysis is finally discussed in the implications section (L303). While I think this is a reasonable hypothesis, it would be strengthened by a more structured set-up connecting morphometrics and dynamics. The full discussion of the morphometrics could be included in the implications. If the authors are making the argument that a lack of vegetation and high low-curvature migration rates leads to channel wandering and limited sinuosity, a side-by-side discussion of channel cutoffs (which can also be influenced by vegetation) has to be included to make the argument convincing.

While the depth of the analysis produces significant results, a few methodological choices can use better justification to strengthen the manuscript:

- There is little explanation as to how or why the data is/is not aggregated. The dataset includes 54 rivers, but all the analyses are completed at the bend-scale. This means that dependent measurements at the bend-scale, are treated as independent measurements throughout all statistical tests (e.g. inferential statistics in the morphometry and the comparison of width-averaged migration rates). While bend-scale measurements have to be used for the curvature analysis, using bend-scale measurements for the WRS and KS tests will lead to artificially low p-values. This is a significant point because all the individual bend-scale measurements are **not** independent measurements and some sort of bootstrapping is essential for correcting for this inherent bias. The authors really have 54 independent rivers and not hundreds to thousands of independent measurements as the data plots make it out to be.
- Why do the authors select the morphometric variables used in the analysis (L93)? There is a good deal of overlap between variables. For example, what is gained by including intrinsic wavelength, cartesian wavelength, and sinuosity?
- What is the PCA analysis contributing to help flesh out the results from inferential statistics? Is the most significant takeaway that only sinuosity and skewness are important (L143)? The authors do include higher order statistics of the variables, and these show up in the results, but are not discussed.
- In the current text, it's unclear whether the mean-normalization (L253) is producing a result that is different than the width-normalization (L216). The authors include reporting the curve-fits for the mean-normalization (L262-L268), but not the width-normalization. It currently reads that the mean-normalization simply reduces the noise in the semi-vegetated rivers. A related question, is the correct interpretation for figure 4c (unvegetated) that the median migration rate for an unvegetated straight reach is faster than the average migration rate for the river?

The figures contain a wealth of information, but they are at times difficult to unpack. A couple of notes on the figures:

Figure 2:

- The "inset" is the last subplot on the figure, but cited first. This should be directly labeled.
- The faint shading that is used in the figure makes it hard to read/interpret. This is most striking in A-D where only the axes appear to have 100% opacity.
- Including all the box plots, probability density plots and exceedance probability plots is not necessary. The overlap in spread between the bend-scale data can be seen in either the points from the box plots or the pdf plots. Depending on how the figure is revised, plotting some version of the river-aggregated data could also be useful.

Figure 3:

- While Figure 3A would be nice to have in the supplement, the sentiment is covered well in the text

already (L135).

- Because you minimize all components other than 1 in the text (L138), the plot between a2 and a3 is superfluous given that you include more PCA plots in the supplement that also don't show vegetation differences. You could move D/G into the supplement, which might allow you to have bigger symbols, etc. There is an argument to be made that B and E are the only important subplots according to how it's discussed in the text.
- There is also overlap between table 1 and the list of variables included in the figure. The variable names in the figure are faint and hard to read. They could be cut.

Figure 4:

- Subplot A is a bit misleading because the data is not aggregated or bootstrapped. The boxplots will be affected by including dependent bend-scale measurements. There could be an opportunity here to show the river-aggregated data as well.
- The kernel density plots are a great way to show the massive amount of bend-scale data. It's too faint as is. The colors need to be more visible

Overall, I thoroughly enjoyed reading the results and I do think they are noteworthy and significant. The analysis between curvature and migration rates across different vegetation is especially novel. With major revisions this will be an important contribution. The major themes in revisions we suggest are to better justify the use of bank-aggregated data across all analysis, streamline the figures, and connect the two analyses within the same discussion and conceptual framework.

Other specific notes:

L74: What is "individual" in this context. This is only point in the paper that there is a discussion about aggregating at multiple scales. There needs to be more detailed description about what each data point represents.

L104-107: Does this mean that the differences between the unvegetated and semi-vegetated rivers are statistically similar? Or did you only test for differences with the vegetated rivers?

L112: Provide a count of unvegetated and vegetated rivers that fall into these categories to back this up. That way the reader doesn't have to go into the tabulated data.

L131: Is cross-correlation the reason some variables were left out? Could swap this and the previous sentence to help clarity. Did the authors use any standardized method to determine which variables to use, i.e., a threshold? Were things like average half sinuosity and half meander intrinsic length not correlated?

L178: Wouldn't this mean that the avulsion timescale would have to approach bend-migration timescales (the time necessary to reach a "mature" bend).

L189: Why are avulsions not a concern if they are mentioned as an important consideration at the migration timescale in the previous section?

L213: This is a fairly novel connection of ideas based on your Geology comment of Sylverster et al. 2019. It may be useful to give just a sentence more detailed description to strengthen this key point past "hydrodynamic nonlinearities".

RESPONSE TO REVIEWERS ON

"Vegetation enhances curvature-driven dynamics in meandering rivers" by Finotello et al. (Manuscript ID: NCOMMS-23-37093)

Please note that in this response, *italics* refer to the text of the reviewers' comments, our detailed replies are in black, and the new text of the revised version is in **bold** font. Line numbers in our replies refer to the revised version of the manuscript.

REVIEWER #1 (REMARKS TO THE AUTHOR):

Overall evaluation: This is an important and informative evaluation of the role of vegetation on meandering behavior. The methodology and analyses are largely clearly explained. Most of the detailed comments can be easily addressed. The issues raised in the general comments are largely beyond the scope of the paper, but the authors should assess whether they have implications for interpretations in the paper.

Thank you for this supportive assessment of our work.

General comments:

The role of sediment supply characteristics in meandering behavior is not discussed in this paper. In this comment I rely on informal observations that may not be universally applicable. Meandering in arid environments seems to occur only when the supplied sediment contains a high fraction of mud relative to sand and gravel. Sand or gravel-dominated channels tend to be braided (this also applies to glacially-fed channels). The "point bars" of these mud-dominated channels are also cohesive. The presence of vegetation in more humid conditions allows stable meandering in sand and gravel-dominated settings which would be braided in more arid environments. Thus the characteristics of sediment supply may be an uncontrolled factor in the present analysis that is indirectly associated with vegetation.

We fully agree with the reviewer that sedimentological differences between rivers in vegetated and non-vegetated (or poorly vegetated) settings are undeniable. As discussed in the Introduction section, it is precisely these sedimentological differences that allow the development of meanders in non-vegetated landscapes in the first place, with single-thread meandering rivers only forming under favorable conditions (i.e., where cohesive agents such as mud provide the necessary bank strength). As stated by the reviewer, meandering rivers exist in a narrow range of bed material and bank strength (see, for instance, Candel et al., 2021 and references therein), and it is unclear whether those conditions would be met in a world fully devoid of land plants. Thus, our analysis best applies to the variety of landscapes – unvegetated to vegetated – found on the modern Earth. Notwithstanding, mudrock (of detrital origin) is common in the Precambrian record and on Mars, where plants never evolved, such that our analysis may also shed light onto fluvial dynamics in the absence of vegetation (e.g., Ielpi et al., 2022; Lapôtre et al., 2019).

The high frequency of chute cutoffs in channels with low density of vegetation may influence average migration rates due to the perturbation of the flow field, which generally results in rapid adjustments of the planform and locally high migration rates. This is very evident in simulations of meandering where neck cutoffs result in fast migration, sometimes with reverse direction of migration. This is largely compensated for in the present study by considering in detail only the low-curvature bends, but could still have some influence on low-curvature bends.

Agreed. We also notice here that neck cutoffs, which are incidentally more widespread in vegetated rivers for the reason discussed at l. 286-292, likely have a stronger influence on enhancing migration rates because they produce higher river curvature than chute cutoffs, and migration rates are positively correlated with higher curvature values in the way we demonstrate. Notwithstanding, we added the following sentence in the revised "Curvature-driven meander dynamics enhanced by riparian vegetation" section:

"In addition, local perturbations of river morphology induced by widespread chute cutoffs in barren

and poorly vegetated settings likely lead to accelerating migration and channel widening both locally and nonlocally^{68,69}, thus further contributing to enhancing m and data scatter at low curvature values that are typical of chute cutoffs. All of these processes are likely to make bank erosion and related river migration less correlated with curvature as the density of riparian vegetation decreases, thus explaining our observations.”

Specific comments:

Line 24: The connotation of “morphodynamic maturity” will likely not have meaning to the reader until it is defined.

To keep the abstract as simple as possible, we have revised the sentence that now reads:

“We show that denser riparian vegetation correlates with enhanced meander sinuosity, skewness, and curvature.”

Line 28-29: It is unclear what is implied by “enhances the proportionality between migration rate and channel curvature”, i.e., a higher correlation coefficient, a higher regression slope, or a higher intercept? It is unlikely to be the latter since the migration rate is lower according to the first clause of the sentence.

We meant regression slope (i.e., the coefficient of proportionality). We have revised the sentence as follows:

“Nevertheless, we find that vegetation demonstrably reduces the rate of river lateral migration, and enhances the coefficient of proportionality between migration rate and channel curvature”

Line 29: The antecedent “effect” is unclear for this sentence. Is it the reduced “rate of river lateral migration”? Also, what is “obscured”? The rate of migration?

We have rephrased the sentence entirely as follows:

“This enhancement is most evident at relatively low curvature and wanes in curvier channels regardless of vegetation density”

Line 31: Eliminate “Hence,”, because this has not been demonstrated in the abstract. The sentence works without this.

Done, thank you.

Lines 44-46: The text leaves unresolved the question of why heterolithic strata are uncommon prior to the development of vascular plants, given that vegetation is asserted to be not necessary for meandering behavior. Is this issue addressed later in the paper?

This aspect, albeit important, is largely beyond the scope of the present work. Although there is an approximate alignment noted between the extended proliferation of land plants during the early to middle Palaeozoic era and the emergence of sedimentological markers associated with lateral accretion in modern meandering rivers, lateral accretion sets in modern meandering rivers in unvegetated landscapes are not heterolithic (Hasson et al., 2023). It is thus possible, if not likely, that many deposits from pre-vegetation meandering streams have remained undetected to date.

Line 62: It is unclear what would be implied by the inverse relationship implied by “vice-versa”. Would meandering somehow lead to development of vegetation?

That is precisely what we meant here. Indeed, Santos et al. (2017) posit that the abrupt and widespread appearance of fluvial deposits, signalling the accumulation of meandering river systems during the Middle Palaeozoic on Earth, was predominantly a consequence of prevailing environmental and tectonic conditions during this period, leading to a global rise in the prevalence of meandering rivers that, in turn, facilitated a conducive environment for the colonization of continents by land plants. This hypothesis has sparked a debate within the scientific community (Davies et al., 2017; Santos, Mountney, Peakall, et al., 2017), and therefore, we believe it is necessary to mention these references in our introduction.

Nonetheless, to avoid confusion, we have removed “vice versa” from the original sentence. The sentence works without this.

Line 78 and lines 418-450: The present authors are probably not responsible for the misnomer “dynamic” in dynamic time warping. Based on the description, this methodology is at best “kinematic” because it is not based on mechanistic, deterministic forward modeling.

The “dynamic” term in DTW refers to the fact that the warping or alignment of two data series is not predetermined but is determined dynamically during the computation. Although we agree with the reviewer that this might seem confusing when used in the context of morphodynamics, DTW is a well-established technique used in the field of pattern recognition, time series analysis, and data mining, and using a different term than what is used broadly in the literature would be even more confusing.

Line 88-89: Because “x” is a scaled variable in the equation, it probably should be designated as “x”.*

Correct. To avoid confusion with $x^*(s^*)$ (mentioned in methods as the Easting coordinate of channel axis centerline), we have changed x to z^* . Thank you for noting this.

Line 91-92: “fairly constant” is not quantifiably interpretable. Does it imply statistical equality of values?

To avoid misinterpretation, we have rephrased the sentence as follows:

“Notably, whereas power-law prefactors (a) differ slightly among distinct vegetation classes, power-law exponents (b) range between 0.98 and 1.01 and are all sub-equal to 1, implying a linear scaling relationship between B^* and all analyzed morphometric descriptors.”

Line 96: Clarify what is meant by [-].

“[-]” denotes dimensionless variables. For consistency with other dimensionless variables mentioned in the text, we have removed it.

Line 102-103: “significant differences” should be “significant differences in morphometric relationships”.

We have revised the sentence as suggested.

Line 132: The criteria for this downsizing is not specified. This is not a crucial issue, but could be mentioned in the supplementary materials or methods.

The reason for this downsizing is that some variables are strongly cross-correlated, which is implicitly mentioned in the previous sentence. Therefore, retaining the entire set of variables results in two (or more) of them having the same effect in terms of data separation in the PC space. This is clear from Supplementary Figure 1, which shows PCA results utilizing the entire set of morphometric variables.

In the main text, we have added a brief explanation regarding the downsizing (following also suggestions from other reviewers) that reads:

“Whereas the resulting dataset consists of 28 variables overall, we performed PCA on a subset of 17 variables (Fig. 3) that succinctly captures the variation in the original data while avoiding redundancy from cross-correlations.”

Line 146: Here again, “maturity” of bends is mentioned without precise definition.

We rephrased the sentence as follows:

“Higher χ , larger C , and lower A values are typically associated with late-stage meander growth^{40,41}, suggesting that denser riparian vegetation correlates with the continued growth of meanders until they reach morphodynamic maturity. At this stage, bends become highly sinuous and are more likely to shortcut themselves through neck cutoff⁴⁰, representing the breaching of the

narrow floodplain limb separating two adjacent bends that marks the endpoint of meander evolution (Fig. 1D).”

Line 170: Is “impulsive” a well-recognized descriptor of sediment transport behavior?

To avoid confusion, we have changed the sentence as follows:

“Flashier hydrological and sediment-transport regimes in arid and semi-arid settings may also further facilitate meander chute cutoffs, compounding the priming effect of reduced riparian vegetation density.”

Lines 183-186: I’m not sure why the effect of vegetation on chute cutoffs is a “confounding” factor and makes it challenging to infer causation. Cutoffs are probably the dominant factor in explaining the lower average sinuosity in arid-zone channels due to sparse floodplain vegetation.

We do not fully understand this comment. Lines 183-186 in the original manuscript read: “(...) more frequent avulsions in modern unvegetated and sparsely vegetated settings are a confounding factor that makes it challenging to infer causation between vegetation and meandering river planforms.” We believe this is correct because avulsion (not cutoff) is implied here as a confounding factor. The reviewer is correct in stating that meander cutoff can be a dominant factor in explaining the lower sinuosity of unvegetated meandering rivers. However, the significant disparity in avulsion regimes between modern unvegetated and vegetated meandering rivers makes it impossible to ascribe lower sinuosity to a higher chute cutoff frequency alone. This is because avulsion effectively limits the development of meander sinuosity by shifting the river course and causing the sinuosity development process to begin anew.

Line 211: something is missing in the phrase “pointing to a strong first-order control of C meandering river morphodynamics”

Yes, thank you for noting that. The correct sentence is:

“(...) pointing to a strong first-order control of C on meandering river morphodynamics.”

Line 217 and Supplementary Material 2: It is unclear what is meant by “correcting migration rates by the average spatial lag”. The effect of the lag is clear in Supplementary Figure 10. Is the lag used to “correct” the data the average of the individual lags applied to the reach as a whole, or is the lag determined and applied to each bend individually? Parenthetically, the term “corrected” should probably be “normalized” or “adjusted”.

At line 217, we rephrased the sentence using 'adjusts' instead of 'correct' and specified that the spatial lag is taken as the reach-average value. This is in agreement with previous studies (Sylvester et al., 2019b) that demonstrated this procedure to be appropriate since the lag exhibits limited variability along any given river reach. The revised version of the sentence reads as follows:

“In order to explore the effect of vegetation on curvature-driven meander dynamics, we derived the relationship between dimensionless curvature (C) and width-adjusted migration rate (M_R) for each river in our dataset, further adjusting migration rates by the reach-average spatial lag (Δ_{CR}^* [m]) between local maxima in channel curvature and migration rate^{37,44,51} (Supplementary Method 2).”

In the revised version of “Supplementary Method 2” we also better specify the above point. The revised Supplementary text reads:

“Hence, the curvature-migration lag is not necessarily constant but rather a function of the local morphodynamic regime and related meander features². Moreover, the lag is likely to vary between different river systems depending on a range of factors (e.g., underlying geology, sediment characteristics, and environmental conditions). This notwithstanding, previous analyses suggested that the along-channel distance between peak curvatures and migrations is fairly consistent across a wide range of climates and geological settings, typically attaining values ranging between 2 to 3 channel widths^{33,39,40,92}. Our data confirm the above. It is clear from measurements of local migration rates in time-lapse satellite images that the pattern of river migration rate closely follows that of the local curvature, with a roughly constant phase lag between the two (see an example from the actively migrating Rio Horton in Supplementary Fig. 11). Moreover, as illustrated by Supplementary Fig. 14, the reach-averaged curvature-migration lag (Δ_{CR}^* [-]) consistently scales with river width across a

wide range of environmental and climatic settings (as typified by distinct riparian vegetation density), with values just slightly larger than two channel widths. Thus, in order to account for the spatial lag between curvature and migration maxima, in our analyses we first computed M^* and C^* separately (see Materials and Methods in the Main Text), and then shifted the M^* signal upstream by a length Δ_{CW} , so that we could study the M^* vs. C^* functional relationship in a physically sound fashion.”

In Supplementary Figure 13 all the lags are positive, presumably implying a sub-resonant regime. Was this true for all rivers studied?

The dominant morphodynamic regime can be observed based on the distribution of meander asymmetry, as reported in Figure 2J. In the revised version of Fig.2, we have also included data points related to individual meandering river reaches on top of the data concerning individual meander bends (which were already reported in the early version of the manuscript). It is immediately apparent that asymmetry index values are consistently negative on average, both when considering individual river reaches and the distribution of individual meander data, regardless of the presence or absence of riparian vegetation. This indeed points to a sub-resonant morphodynamic regime in all analyzed rivers, although the presence of individual, downstream-skewed bends indicative of local super-resonant dominance is also observed. This comes as no surprise since most meandering rivers are known to fall within the sub-resonant domain.

Materials and Methods: Extraction and analysis of meandering river planforms: As is acknowledged in the discussion care needs to be taken to remove “noise” related to the digitizing procedure in determining the generalized planform. The discussion is generally clear about the procedures, but some detail is lacking due to reliance on previous studies. One of the most critical steps in segmenting natural channels is defining the inflection points. The issue is illustrated in Supplementary Figure 10, where near the north arrow there is a slight inward inflection of the adjacent long bend. If the plotted centerline between iv and v is correctly plotted there should be two intermediate inflection points and one half-bend that are not evident in part B of this figure. This issue only becomes important if including those inflection points would significantly affect the subsequent analyses. Have the authors, or previous studies, evaluated the influence of planform generalization on resulting morphometric parameters?

The absence of an inflection point (two, in fact, since a complete yet short meander bend is defined) between inflections 'iv' and 'v' in SI Fig. 10 is attributed to the low-pass filtering of the curvature signal, a point that we agree was not sufficiently emphasized in the original manuscript. This filtering, which compounds the filter and smoothing procedure (via cubic spline) applied to river centerlines, is necessary to eliminate remaining noise in the curvature signal and prevent the identification of spurious inflection points, as correctly anticipated by the reviewers based on SI Fig. 10. Note that in this context, we use the term 'spurious inflections' to refer to zero-crossing points in curvature signal describing short meander bends with extremely low sinuosity values. These spurious inflections are extremely common in double-headed meander bends characterized by two local maxima of curvature; multiple examples can be found in SI Fig. 10. Specifically, the river planform geometry shown in Fig. 10 was characterized by the presence of two zero-crossing points in the curvature signal between inflections 'iv' and 'v.' These zero crossings in the original curvature signal, which described a pseudo-meander with an intrinsic length approximately equal to 1 channel width and a sinuosity <1.01 (which is why we call it a “pseudo-meander”), were automatically smoothed away by our filtering procedure. It is important to note here that no custom-user correction was applied, ensuring homogeneity in data analyses for all the river centerlines in our dataset. Since river centerlines were resampled at a standard resolution of one-tenth of the channel average width, our filtering procedure remains consistent across channel sizes and does not affect our results (as spurious inflections are discarded or retained based on an automatic size-independent procedure). This consistency effectively filters out size-dependent oscillation in meander morphometrics, a long-known issue that has been recently addressed by Stanislawski et al., (2023) with respect to meander sinuosity, for example. We also emphasize that double-headed meander

bends are the exception rather than the norm along a given meandering river reach and are not expected to critically impact the results reported in our study.

To better clarify this point, we have updated the “Extraction and Analysis of Meandering River Planforms” section in Material and Methods (main text) as follows:

“To isolate individual meander bends we used a semi-automated procedure based on the computation of local channel-axis curvature C^* ($[m^{-1}]$) (Supplementary Fig. 11). Specifically, for any given centerline point $\{x^*(s^*), y^*(s^*)\}$, we computed $C^* = d\theta(s^*)/ds^*$, where s^* ($[m]$) denotes the channel centerline curvilinear coordinate, $\{x^*(s^*), y^*(s^*)\}$ represent the coordinates of an arbitrary axis point in a Cartesian reference system, and $\theta(s^*)$ is the angle formed by the tangent to the channel axis and an arbitrarily fixed direction^{33,37}. After computing C^* , a Savitzky–Golay low-pass filter with a fixed polynomial order of 3 and a frame length of 21 centerline points was applied to further smooth noise in the original C^* signal. Subsequently, half (full) meander bends were identified as the portion of the channel between two (three) consecutive inflection points (i.e., points where $C^*=0$). Meander apexes were also identified as points corresponding to local curvature maxima (Fig. 1C and Supplementary Fig. 11). We note that the low-pass filtering of C^* signal resulted in the automatic deletion of some “spurious” inflection points delimiting meander bends characterized by limited length and sinuosity, which are typical in double-headed meander bends featuring multiple local maxima in the C^* signal (Supplementary Fig. 11).”

REVIEWER #2 (REMARKS TO THE AUTHOR):

This is a very timely work that reveals the role of vegetation in meandering river development. Multiple controls have been previously proposed for the long-standing debate about the necessary conditions for meandering river development. Among these controls, vegetation has stirred up a lot of discussions recently, but how specifically does vegetation affect the morphology and dynamics of meandering river have not been systematically explored in sufficient details until this current manuscript. This work documented a list of key geometric attributes of meandering river planform morphology across a wide range of riparian vegetation densities. Statistical analyses of these attributes reveal that the impact of vegetation on meandering morphology and dynamics is subtle but significant. Moreover, the author found the range of curvature that distinct vegetation control operates. These results have important implications for the role of riverine environments in global carbon cycle and reconstruction of the morphology and dynamics of ancient rivers on Earth and Mars.

Thank you for this supportive assessment of our work.

There is one missing aspect of the dataset analyzed that could potentially hinder the application of the results from this study. It seems that the rivers in the analyzed dataset collected by the authors are all small to intermediate-sized rivers, and most large lowland rivers are not included. This raises the question whether the findings of the study apply to large lowland meandering rivers as well. I speculate that the impact of vegetation on large rivers is limited as the flow depth of these rivers extends far below the roots of vegetations. This does not negate the results of this study but suggests that there might be a river size limitation on the role of vegetation on meandering river morphology and dynamics. In order to maximize the application of this work, I would suggest the author to consider the following:

(1) Include some large rivers in the dataset. Whether the large lowland rivers follow the trends demonstrated in this study will be useful information. If they do, then it supports that the impact of vegetation is also important for large rivers. If not, there might be a river-size limit for vegetation to affect the morphology and dynamics of meandering river. This is not to underrate the importance of small- to intermediate sized rivers, as they comprise a large portion of global rivers.

(2) Characterize the size of the rivers in the dataset in terms of slope and discharge and provide the overall range/distribution of these metrics. This will help to put the size of the rivers in the dataset into a spectrum so that in case no large river will be included in the analysis, at least it shows the slope and discharge range that the current results are applicable to.

We thank the reviewer for this important comment, which highlights the need for a deeper discussion of our selection criteria. The present dataset includes vegetated meandering rivers that are up to ~1 km wide; while we'd certainly like to include more, larger rivers are inherently few. Furthermore, we only consider (1) single-thread meandering rivers that are (2) not significantly affected by human activity, and (3) are sufficiently far away from the coastline in order not to be affected by backwater and/or tidal effects. These criteria significantly restrict the pool of case studies that are suitable for expanding our dataset. Finally, we only considered reaches containing at least 35 consecutive bends, so that a sufficiently long train of meanders can be analyzed to ensure data homogeneity among distinct fluvial settings. We could thus not include stretches of >1 km wide meandering rivers comprising fewer than 35 meander bends in our analyses. Together, we believe that these criteria leave the pool of available candidate rivers to exactly zero. To check this, we used the global dataset provided by Frasson et al. (2019), filtering out river stretches that i) are narrower than 1050 meters; ii) belongs to lake/reservoirs; iii) have an overall sinuosity lower than 1.2 (the filtered Frasson et al.'s database can be accessed, for the reviewer benefit, at the following link:

<https://drive.google.com/drive/folders/1yKThmeFQT37QE-hfHp7leUAeNvNi1ALB?usp=sharing>).

It clearly emerges that the only handful of large rivers that meet our selection criteria are not free from human influences (e.g., the lower stretch of the highly engineered Mississippi River), and therefore cannot be included in our dataset. In summary, our dataset includes the largest single-

thread, sufficiently long, undisturbed vegetated river stretches we are aware of. In the revised version of our manuscript, we now discuss those selection criteria in more detail. Specifically, the revised “Material and Methods - Extraction and Analysis of Meandering River Planforms” section now includes:

“To obtain planform morphometrics of individual meandering reaches, we first hand-digitized riverbank lines in the QGIS environment based on freely available georeferenced aerial and satellite images. The latter include several different products available through Google Earth Pro, SASPlanet, the U.S. Geological Survey Earth Explorer portal, and the QuickMapService QGIS plugin. Additionally, we included the data collection provided by Sylvester et al.³⁷ for vegetated meandering rivers in the Amazon basin in our dataset (Fig. 1A).

In order to ensure data homogeneity among distinct fluvial settings, we only considered single-thread meandering river reaches that i) contain at least 35 consecutive bends, such that a sufficiently long train of meanders can be analyzed; ii) are found sufficiently far away from the coastline in order not to be affected by backwater and/or tidal effects; iii) have not been significantly modified or engineered by humans.“

As a side note, we emphasize that the lack of large unvegetated streams on modern Earth cannot be used to infer that vegetation enables larger meandering systems because vegetation is everywhere in present-day Earth except for small arid endorheic basins draining relatively small catchments.

In terms of bed slope and flow discharge, the other two parameters requested by the reviewer, we note that:

- Not all the analyzed rivers are gauged (most are not, in fact, especially smaller and poorly vegetated ones); since river width (W) is known to scale with formative discharge (Q), we could use known global relationships to derive Q from W , but this would add nothing to the proposed analyses currently carried out based on W .
- Slope is perhaps the easiest parameter to estimate, which can be done, for example, based on remotely sensed SRTM data. However, critical changes in river size (with W spanning 4 orders of magnitude in our dataset) combined with the fixed resolution of SRTM data would result in biased estimates of slope depending on river size. Furthermore, we note that slope (like discharge) would be somewhat redundant with width (for given bed and bank materials) as we only consider self-formed, alluvial rivers.

Some minor points for the author to consider:

Is there a threshold curvature used for identifying cross-over point? For example, there seems to be low curvature bends that are part of a compound bend but not considered as a bend defined by cross-over point in figure 1C (e.g., upper section of Humboldt river below the 7th cross-over point, and mid-section of the Rio Horton river below the 13th cross-over point). It seems that the current method skips some of the low curvature bends.

No, there is no pre-defined curvature threshold used for identifying cross-over (i.e., inflection) points. However, a low-pass filter is applied to the curvature signal that automatically filters out some “spurious” inflections as discussed above in our response to Reviewer 1’s last comment.

Specific comments:

Line 71: what about different types of vegetation, which may contribute to the spread of data in the analysis?

Different types of vegetation could clearly have different impacts on river morphodynamics, due to, for instance, varying rooting depths. However, rooting depth is not just a function of vegetation species but also depends on local factors such as the elevation of the water table and soil properties, for example. Assessing the specific types of vegetation species is impractical based on remotely sensed NDVI data alone (multispectral data at proper resolution would be needed, together with ground thrusting surveys). In addition, different vegetation species suited to distinct climate settings

might have similar rooting depths, which further complicates possible detailed analyses. The latter aspect is beyond the scope of the present work. Notwithstanding, we agree with the reviewer that not only riparian vegetation density but also the type of vegetation can affect both river morphology and dynamics. However, we shall also note that denser riparian vegetation typically (i.e., higher NDVI values, say $NDVI > 0.4$) consists of trees and arborescent shrub species, with poorly vegetated settings ($0.2 < NDVI < 0.4$) featuring mostly shrubs and ground covers. In this way, a transition from high to low NDVI values also denotes broad changes in plant taxonomy. A discussion on possible effects related to river size and vegetation type, not necessarily exhaustive, is reported in the text:

“Plant density, as well as channel width and depth relative to plant stem diameter and rooting depth^{14,25,61–63}, have been shown to influence channel dynamics. In sparsely vegetated landscapes, shrubby arborescent, drought-resilient plants will encroach into the moister and nutrient-rich thalweg zone of ephemeral channels, where groundwater is more accessible^{17,25}. Especially in rivers narrower than 10 m, such clusters of in-channel vegetation can enhance local scour and bank erosion, disrupt curvature-induced helical flow that is necessary to sustain lateral migration, and limit meander growth by facilitating braiding and meander chute cutoffs^{25,62} (formed when the river cuts a new bypass channel through its own point bar⁶⁴; see Supplementary Fig. 6). In contrast, deep-rooted trees can sustain large single-thread meandering channels (width $>10^2$ m)^{25,63} in densely forested floodplains, aiding flow confinement through enhanced bank-erosion resistance and facilitating the continued development of highly sinuous meander bends until neck cutoff. On average, larger rivers are associated with higher NDVI values (Supplementary Fig. 7), but our analysis of normalized morphometric characteristics shows that the effects of vegetation regime on meandering dynamics are independent of channel size (Fig. 2A-D).”

Line 177 “low-gradient terminal fluvial fans...”

Could you provide the actual slope range for the studied rivers.

We have revised the sentence as follows:

“Besides, meandering streams in modern barren environments are found almost exclusively along low-gradient (slope $\sim 10^{-5}$ - 10^{-4}) terminal fluvial fans where relatively high rates of vertical aggradation increase the frequency of river avulsions^{5,21,25,80}”
(see also our detailed response to a previous comment).

Line 212 “reaches ($C < 0.3$)...” How is this threshold value of curvature determined? Seems the curvature for peak adjusted migration rate for semi-vegetated and unnegated rivers is closer to 0.2 (Fig. 4B, C).

The threshold value of curvature we used is based on both the results we present in our manuscript and earlier literature. Specifically, beginning with the seminal work of Edward Hickin and Gerald Nanson on the Beatton River (British Columbia, Canada) (Hickin & Nanson, 1975), it has been recognized that migration rate maxima occur for a value of bend radius over width $2.5 < R^*/W^* < 4$ (see also Blanckaert, 2011 and Hooke, 2013, among others). Such a range of bend radii, which corresponds to a normalized channel curvature $0.25 < C^* \cdot W^* < 0.4$, is not incidental, as it depends on strong hydrodynamic nonlinearities arising in sharp bends (e.g., Blanckaert, 2009, 2010; Blanckaert & de Vriend, 2005; Finotello et al., 2019; Ottevanger et al., 2012). In summary, although there isn't a specific curvature value associated with the growth of such nonlinearities and the related saturation of migration rates, we used $C=0.3$ as a typical value to avoid constantly referring to a range of critical $C^* \cdot W^*$ values.

We have also specified this in the revised manuscript:

“This proportionality typically only holds for mildly curved channel reaches (i.e., for width-adjusted curvature values smaller than 0.25-0.5, taken here as $C < 0.3$ for convenience) and breaks down at high curvatures ($C > 0.3$) where M_R saturate due to the growth of hydrodynamic nonlinearities that effectively limit bank erosion, such as saturation of centrifugally driven secondary flows, enhanced secondary outer bank cells, and flow separation at the outer bank^{1,38,46,53,54} (Supplementary Method 2).”

Line 236 “morphodynamic feedbacks that arise...”

What specific feedback this is referring to?

This is explained in the next paragraph:

“Plant density, as well as channel width and depth relative to plant stem diameter and rooting depth^{14,25,61–63}, have been shown to influence channel dynamics. In sparsely vegetated landscapes, shrubby arborescent, drought-resilient plants will encroach into the moister and nutrient-rich thalweg zone of ephemeral channels, where groundwater is more accessible^{17,25}. Especially in rivers narrower than 10 m, such clusters of in-channel vegetation can enhance local scour and bank erosion, disrupt curvature-induced helical flow that is necessary to sustain lateral migration, and limit meander growth by facilitating braiding and meander chute cutoffs^{25,62} (formed when the river cuts a new bypass channel through its own point bar⁶⁴; see Supplementary Fig. 6). In contrast, deep-rooted trees can sustain large single-thread meandering channels (width>10² m)^{25,63} in densely forested floodplains, aiding flow confinement through enhanced bank-erosion resistance and facilitating the continued development of highly sinuous meander bends until neck cutoff. On average, larger rivers are associated with higher NDVI values (Supplementary Fig. 7), but our analysis of normalized morphometric characteristics shows that the effects of vegetation regime on meandering dynamics are independent of channel size (Fig. 2A-D).”

Line 249 “...independent of channel size”

Again, what about large rivers.

Please refer to our detailed response to a previous comment.

*Chenliang Wu
Postdoc Researcher
Tulane University*

References

I. Zhu, L., Chen, D., Hassan, M. A. & Venditti, J. G. The Influence of Riparian Vegetation on the Sinuosity and Lateral Stability of Meandering Channels. Geophysical Research Letters 49, e2021GL096346 (2022).

REVIEWER #3 AND #4 (REMARKS TO THE AUTHOR):

Review of Vegetation Signature on River Meander Morphology and Dynamics by Finotello et al. submitted to Nature Communications

Finotello et al. provide a comprehensive analysis to show how vegetation influences river meandering. The manuscript is well written and the authors should be applauded for their detailed science and valuable data its created. The work is significant. The authors find that rivers with unvegetated and vegetated banks have different planform sinuosities and bend asymmetries suggesting that bends don't reach maturity in unvegetated settings. They also use a novel analysis and show that while rivers with all types of bank vegetation have consistent curvature vs. migration rate relationships, unvegetated rivers have faster bank migration rates at low curvature. The methodology and data analyses are very sound, and the results are of broad interest to fluvial geomorphologists and geologists that keep up with the debate on the role of vegetation in river meandering.

Thank you for this supportive assessment of our work.

With that said, there are a few overarching difficulties I had with the present version of manuscript that include:

- 1) more discussion connecting the morphometric and migration rate analyses;*
- 2) a need to better justify methodological choices;*
- and 3) figure simplification and readability.*

We are thankful for the constructive review from the present reviewers. We addressed all suggestions in full, as detailed below.

There are two related, but separate analyses included in the manuscript. The first half of the article focuses on a morphometric analysis of river planform. They utilize and report on two different statistical approaches (population inference and PCA). The second half of the paper is a novel test of existing empiricism and theory that for the first time in my knowledge, looks at curvature and meander migration for both unvegetated and vegetated rivers. However, there is little structure connecting these two analyses.

Some notes on this:

- the “Other Impacts on Planform Morphology” section (L150) is indicative of the difficulty the text had in bridging the two analyses. I found this section out of place as it was a discussion describing drivers of planform morphology that aren't explicitly vegetation without a clear section devoted to discussing the role of vegetation itself on planform morphology.*
- further, the opening of the “Meandering river dynamics” section (L189) is too generic and can use more justification as to why were are looking at morphometrics and dynamics in the same analysis. Instead, the paragraph (L189-L203) spends valuable text comparing to previous literature, which while it is great to see that it lines up, could be used to tie the analyses together.*
- the primary connection between the two analysis is finally discussed in the implications section (L303). While I think this is a reasonable hypothesis, it would be strengthened by a more structured set-up connecting morphometrics and dynamics. The full discussion of the morphometrics could be included in the implications. If the authors are making the argument that a lack of vegetation and high low-curvature migration rates leads to channel wandering and limited sinuosity, a side-by-side discussion of channel cutoffs (which can also be influenced by vegetation) has to be included to make the argument convincing.*

In hindsight, we realize that the connection between morphometrics and dynamics was not as compelling as we wish it were. To strengthen that connection, we restructured the manuscript significantly; notably, through a discussion of results from both morphometric and dynamic analyses, followed by a discussion of their joint implications in the 'Discussion and Implications'.

While the depth of the analysis produces significant results, a few methodological choices can use better justification to strengthen the manuscript:

*• There is little explanation as to how or why the data is/is not aggregated. The dataset includes 54 rivers, but all the analyses are completed at the bend-scale. This means that dependent measurements at the bend-scale, are treated as independent measurements throughout all statistical tests (e.g. inferential statistics in the morphometry and the comparison of width-averaged migration rates). While bend-scale measurements have to be used for the curvature analysis, using bend-scale measurements for the WRS and KS tests will lead to artificially low p-values. This is a significant point because all the individual bend-scale measurements are *not* independent measurements and some sort of bootstrapping is essential for correcting for this inherent bias. The authors really have 54 independent rivers and not hundreds to thousands of independent measurements as the data plots make it out to be.*

This is a good point. We first note that not all analyses were conducted at the bend scale. Specifically, morphometric variables are reported both as reach-averaged and bend-scale values. For instance, in Fig. 2, boxplots represent bend-scale morphometrics, while exceedance probability plots denote reach-scale distributions of morphometric variables (i.e., we had indeed 54 individual plots, one for each river in our dataset). Accordingly, whereas WRS tests were performed with reference to bend-scale data, KS tests referred to probability distributions of reach-scale morphometrics. In this way, we demonstrate that reported differences are statistically significant both with reference to individual bends and river-aggregate planform characteristics of meander bends. However, it is clear from the reviewer's comment that this distinction was not clear in the original version of the manuscript. To address this issue, we now included, in panels A-D of Fig. 2 and in each boxplot within the same figure, data points of reach-averaged morphometric values for every river in our dataset, in addition to bend-scale data. Furthermore, we now also perform WRS tests considering aggregate river data instead of bend-scale data. The updated results of these tests are now included in both the text and the table. It is important to highlight that considering aggregate data yields minimal differences compared to the previously reported results; specifically, no statistically significant changes are observed in terms of differences among the considered morphometric variables (i.e., aggregating data does not significantly impact our result). The revised version of the text reporting the results of statistical tests now reads:

“Two-sample Wilcoxon Rank Sum (WRS) and Kolmogorov-Smirnov (KS) tests, performed at a standard 5% significance level on river-aggregated data, highlight statistically significant differences in morphometric relationships between vegetated rivers (used as the control group) and both unvegetated and semi-vegetated rivers. The KS tests all reject the null hypothesis that the morphometries of unvegetated and semi-vegetated rivers come from the same distribution as those of vegetated rivers. Additionally, the WRS tests reject the null hypothesis that the morphometric parameters of both unvegetated and semi-vegetated rivers have the same median as vegetated rivers in all cases except for the meander amplitude in unvegetated rivers and meander wavelengths and curvature in semi-vegetated rivers.

Compared to their vegetated counterparts, rivers in unvegetated settings typically feature longer meander wavelengths and larger radii of curvature relative to channel width, with lower meander sinuosity and width-adjusted curvature (Fig. 2E,F,H,I,K). Specifically, while in vegetated rivers 26% of meanders have a sinuosity $\chi > 2$ and 13% have a curvature $C > 0.3$, these percentages decrease to 7% and 5%, respectively, for unvegetated rivers. Vegetation also appears to correlate with bend asymmetry: meanders in vegetated and semi-vegetated rivers are more strongly upstream-skewed (i.e., lower values of α) whereas unvegetated rivers host more symmetric bends on average (Fig. 2J). Whereas these differences emerge when data are treated as ensemble averages (i.e., binned over the entire data set), distributions associated with individual rivers display a wide dispersion (Fig. 2E-K). Thus, meander morphologies overlap considerably among different vegetation classes.”

In addition, we eliminated histogram plots of morphometric parameter distributions (previously reported in the form of aggregated plots for each vegetation class) and changed colors to reduce excessive fainting.

• *Why do the authors select the morphometric variables used in the analysis (L93)? There is a good deal of overlap between variables. For example, what is gained by including intrinsic wavelength, cartesian wavelength, and sinuosity?*

The reviewer is correct that not all variables are fully independent. However, as suggested by previous studies (e.g., Finotello et al., 2020; Frascati & Lanzoni, 2009; Howard & Hemberger, 1991), the functional relationships between some of these variables are nontrivial such that no individual variable alone suffices to completely characterize meander bend geometry or, more importantly, to discern any morphometric differences in meander bend subgroups. For other variables, that functional relationship is linear and simple (e.g., wavelength and sinuosity). However, including both greatly facilitates the discussion of our results, as well as allows us to explicitly define the meander size (relative to channel width) producing a given sinuosity value. Furthermore, including all possible morphometric variables enables a direct comparison with previously published literature, broadening the relevance of our work in the context of meandering river research.

L131: Is cross-correlation the reason some variables were left out? Could swap this and the previous sentence to help clarity. Did the authors use any standardized method to determine which variables to use, i.e., a threshold? Were things like average half sinuosity and half meander intrinsic length not correlated?

What is the PCA analysis contributing to help flesh out the results from inferential statistics? Is the most significant takeaway that only sinuosity and skewness are important (L143)? The authors do include higher order statistics of the variables, and these show up in the results, but are not discussed.

As noted in our previous response, many variables are indeed cross-correlated to some extent for purely geometric reasons (and so are most of the variables we retain for the analyses presented in Fig. 3 as well as in SI). This is overall consistent with the use of PCA, which is typically used for dimensionality reduction and summarizing data. Again, we note that this is a standard procedure in the river meander literature, both in terms of methodology and considered morphometric variables (Bogoni et al., 2017; Finotello et al., 2020; Frascati & Lanzoni, 2009; Howard & Hemberger, 1991). The use of PCA not only allowed us to derive critical information regarding differences in meandering river planforms as a function of vegetation (impractical using univariate statistics alone, as we demonstrate), but also aligns our study with previous research wherein it was shown that morphological variations between freely meandering streams can be resolved by morphometric parameters related to wavelength, skewness, sinuosity, and curvature of meander axis (Howard & Hemberger, 1991 being the seminal work in this respect).

Considering that PCA performed on ensembles of morphometrics different from those reported in the main text (SI Fig. S1 and S2) holds similar results, we can conclude that data separation is predominantly driven by higher meander sinuosity, curvature, and degree of upstream skewness in vegetated rivers. This however doesn't mean that only sinuosity and skewness are important, but it rather indicates that these are the variables that explain most of the variance in the dataset.

We shall also notice here that higher-order statistics are included because i) they can be relevant in driving data separation at times and ii) including them is standard practice in the literature (see refs. cited above and also Appels et al., 2008). However, discussing high-order statistics would add little to the reported results in the text at the cost of unnecessarily complicating the reading (besides the fact that the meaning of high-order statistics is debatable *per se*; see for instance Balanda & Macgillivray, 1988; Darlington, 1970; Moors, 1986; Ruppert, 1987; Zhiqiang et al., 2008). Hence, we prefer not to discuss their implications in detail, especially considering that a clear separation of data as a function of vegetation class does not exist (rather, there is a continuous transition as we demonstrate) and that, even if there was a separation, the discrepancy in river avulsion tendency as a function of vegetation would confound it anyway.

Following the reviewer's suggestion, we have modified the sentence in L.131 of the original manuscript as:

“Following a standard approach in river morphodynamics^{33–35,39}, we applied Principal Component Analysis (PCA) (Supplementary Method 1) to a set of distinct morphometric variables derived from river-aggregated distributions of meander planform features (Table 1 and Material and Methods). As demonstrated by Howard and Hemberger³⁵, morphological variations among freely meandering streams can be resolved by morphometric parameters related to meander wavelength, skewness, sinuosity, and curvature. Hence, we considered the first- to fourth-order statistics of the probability distributions of meander sinuosity (χ), asymmetry (\mathcal{A}), and both width-adjusted curvature (\mathcal{C}) and intrinsic wavelength (\mathcal{L}).

All variables were computed for both half and full meander bends (Table 1 and see Material and Methods), except for \mathcal{C} , for which we considered the reach-averaged values. Whereas the resulting dataset consists of 28 variables overall, we performed PCA on a subset of 17 variables (Fig. 3) that succinctly captures the variation in the original data while avoiding redundancy from cross-correlations.”

• *In the current text, it's unclear whether the mean-normalization (L253) is producing a result that is different than the width-normalization (L216).*

The authors include reporting the curve-fits for the mean-normalization (L262-L268), but not the width-normalization.

It currently reads that the mean-normalization simply reduces the noise in the semi-vegetated rivers.

The main effects of mean-normalization are twofold: i) it reduces noise in the semi-vegetated data; and ii) it better highlights the plateau in migration rates for vegetated rivers beyond the $C=0.3$ threshold. Aside from the above, mean normalization has limited impact on reported C vs. M relations. Still, we deem mean-normalization to be necessary to avoid systematic data bias toward highly mobile river reaches. This latter point was not well conveyed in the original manuscript. The revised version of the “Curvature-driven meander dynamics enhanced by riparian vegetation” paragraph now reads:

“To investigate variations in channel mobility while avoiding highly mobile rivers to systematically bias our data, we further normalized M_R by dividing it by the average migration rate of the corresponding river reach ($\overline{M_R}$) (Fig. 4C). Importantly, this procedure also allows for filtering out spurious correlations due to inherent hydrological and sedimentological variability (e.g., flow intermittency and changes in bank erodibility) broadly related to site-specific climatic and environmental conditions.”

Please note that curve fits were reported only for mean-normalized data and only for the $C < 0.3$ data range. We feel this is a sensible decision to avoid excessively lengthening the text, especially since all fits are reported (along with p-values and R-squared coefficients) in Figure 4B,C.

A related question, is the correct interpretation for figure 4c (unvegetated) that the median migration rate for an unvegetated straight reach is faster than the average migration rate for the river?

Assuming that 'straight' corresponds to $C < 0.3$ (although we note that $C = 0.3$ already represents a significant curvature), we obtain, for the unvegetated river considering ensemble data (i.e., all reaches on record), an overall width- and mean-normalized average migration rate of $m_{\text{avg}} = 1.2980$. However, the median migration rate, considering only data with $C < 0.3$, is equal to $m_{\text{straight}} = 0.9902$ (hence smaller than m_{avg}). Even if we restrict our analysis to include only quasi-straight reaches, defined as reaches where $C < 0.001$, we obtain $m_{\text{straight}} = 0.9805$, which is still slower than the average migration rate for the river.

The figures contain a wealth of information, but they are at times difficult to unpack. A couple of notes on the figures:

Figure 2:

- *The “inset” is the last subplot on the figure, but cited first. This should be directly labeled.*
- *The faint shading that is used in the figure makes it hard to read/interpret. This is most striking in A-D where only the axes appear to have 100% opacity.*

• Including all the box plots, probability density plots and exceedance probability plots is not necessary. The overlap in spread between the bend-scale data can be seen in either the points from the box plots or the pdf plots. Depending on how the figure is revised, plotting some version of the river-aggregated data could also be useful.

We modified Figure 2 as shown above in our reply to a previous comment. Regarding the inset, we believe it is appropriate to leave it as it is (unlabelled) precisely because, for graphic reasons, it needs to be placed in the bottom right corner and at the same time, it is the first one mentioned in the text

Figure 3:

• While Figure 3A would be nice to have in the supplement, the sentiment is covered well in the text already (L135).

• Because you minimize all components other than 1 in the text (L138), the plot between a_2 and a_3 is superfluous given that you include more PCA plots in the supplement that also don't show vegetation differences. You could move D/G into the supplement, which might allow you to have bigger symbols, etc. There is an argument to be made that B and E are the only important subplots according to how it's discussed in the text.

• There is also overlap between table 1 and the list of variables included in the figure. The variable names in the figure are faint and hard to read. They could be cut.

Panel A has been moved to the Supplementary Information (SI), and the list of variables has been removed from Fig. 3 as suggested. We agree that panels B and E are the most important ones. However, we prefer to keep D/G in place because it better conveys the message that $PC > 3$ are not important (especially now that we moved the eigenspectrum to SI). Additionally, this is standard practice in recent literature (Bogoni et al., 2017). Note that, having moved Panel A to the Supplementary Information and removed the inset, we have also enlarged the remaining panels to improve readability as requested.

We also modified the colors and shading of panel A-C for consistency with the revised Fig. 2.

Figure 4:

• Subplot A is a bit misleading because the data is not aggregated or bootstrapped. The boxplots will be affected by including dependent bend-scale measurements. There could be an opportunity here to show the river-aggregated data as well.

• The kernel density plots are a great way to show the massive amount of bend-scale data. It's too faint as is. The colors need to be more visible

We modified Fig. 4 as suggested, updating the boxplots using individual-river data (plotted on top of bend-scale data), changing colours for consistency with other figures, and reducing faint shading in KDE plots to make colours more visible.

Overall, I thoroughly enjoyed reading the results and I do think they are noteworthy and significant. The analysis between curvature and migration rates across different vegetation is especially novel. With major revisions this will be an important contribution. The major themes in revisions we suggest are to better justify the use of bank-aggregated data across all analysis, streamline the figures, and connect the two analyses within the same discussion and conceptual framework.

We are thankful for these important suggestions which greatly helped us clarify and streamline our manuscript.

Other specific notes:

L74: What is "individual" in this context. This is only point in the paper that there is a discussion about aggregating at multiple scales. There needs to be more detailed description about what each data point represents.

Agreed, and we have done that throughout the manuscript (as well as in Figures) as suggested.

Concerning l.74 in the original manuscript specifically, we have modified the phrasing as follows to avoid any confusion:

“For each class, we analyzed the width-adjusted morphometric properties of meander planforms at both the scale of single bends and individual river reaches, using classical uni- and multivariate statistical methods in the river-meander literature”.

L104-107: Does this mean that the differences between the unvegetated and semi-vegetated rivers are statistically similar? Or did you only test for differences with the vegetated rivers?

We did only test for differences with vegetated rivers (used as the control group).

L112: Provide a count of unvegetated and vegetated rivers that fall into these categories to back this up. That way the reader doesn't have to go into the tabulated data.

The sentence has been revised as follows:

“Specifically, while in vegetated rivers 26% of meanders have a sinuosity $\chi > 2$ and 13% have a curvature $C > 0.3$, these percentages decrease to 7% and 5%, respectively, for unvegetated rivers.”

L178: Wouldn't this mean that the avulsion timescale would have to approach bend-migration timescales (the time necessary to reach a “mature” bend).

Not precisely. The avulsion timescale T_A is defined (*sensu* Jerolmack & Mohrig, 2007) as the time required for the river to aggrade one (or a fraction of) channel depth vertically, whereas the migration timescale T_M is the time required for the river to migrate one channel width laterally. In this sense, to have mature bends, it is required that T_M is sufficiently small compared to T_A . We note here that to have single-thread meandering rivers (rather than anabranching ones), the ratio $M = T_A / T_M$ needs to be $\gg 1$ (see again Jerolmack & Mohrig 2007). All rivers in our dataset fall in this regime, being all single-threaded, and yet the disparity in T_A might still be able to drive critical morphodynamic differences in the way we illustrate.

L189: Why are avulsions not a concern if they are mentioned as an important consideration at the migration timescale in the previous section?

This is because we are now focusing on meander dynamics in between avulsions, which depends on curvature (both local and upstream-aggregated) rather than on meander planform more broadly defined. Specifically, since the $M = T_A / T_M$ ratio (see previous comment) is still large enough even in the unvegetated class to allow for developing single-thread rivers, the migration dynamics (i.e., rate of lateral migration and the relation between migration and curvature) can be studied in details as long as no avulsion occurs during the timeframe used for analysis.

L213: This is a fairly novel connection of ideas based on your Geology comment of Sylverster et al. 2019. It may be useful to give just a sentence more detailed description to strengthen this key point past “hydrodynamic nonlinearities”.

We have modified the sentence as follows:

“This proportionality typically only holds for mildly curved channel reaches (i.e., for width-adjusted curvature values smaller than 0.25-0.5, taken here as $C < 0.3$ for convenience) and breaks down at high curvatures ($C > 0.3$) where M_R saturate due to the growth of hydrodynamic nonlinearities that effectively limit bank erosion, such as saturation of centrifugally driven secondary flows, enhanced secondary outer bank cells, and flow separation at the outer bank^{1,38,46,53,54} (Supplementary Method 2).”

CITED REFERENCES

- Appels, W. M., Hoitink, A. J. F., & Hoekman, D. H. (2008). Planform geometry of peat meanders. *River, Coastal and Estuarine Morphodynamics: RCEM 2007 - Proceedings of the 5th IAHR Symposium on River, Coastal and Estuarine Morphodynamics, 1*, 271–277. <https://doi.org/10.1201/noe0415453639-c34>
- Balanda, K. P., & Macgillivray, H. L. (1988). Kurtosis: A Critical Review. *The American Statistician*, 42(2), 111–119.
- Blanckaert, K. (2009). Saturation of curvature-induced secondary flow, energy losses, and turbulence in sharp open-channel bends: Laboratory experiments, analysis, and modeling. *Journal of Geophysical Research: Solid Earth*, 114(3), 1–23. <https://doi.org/10.1029/2008JF001137>
- Blanckaert, K. (2010). Topographic steering, flow recirculation, velocity redistribution, and bed topography in sharp meander bends. *Water Resources Research*, 46(9), 1–23. <https://doi.org/10.1029/2009WR008303>
- Blanckaert, K. (2011). Hydrodynamic processes in sharp meander bends and their morphological implications. *Journal of Geophysical Research: Earth Surface*, 116(1), 1–22. <https://doi.org/10.1029/2010JF001806>
- Blanckaert, K., & de Vriend, H. J. (2005). Turbulence structure in sharp open-channel bends. *Journal of Fluid Mechanics*, 536, 27–48. <https://doi.org/10.1017/S0022112005004787>
- Blanckaert, Koen, & De Vriend, H. J. (2003). Nonlinear modeling of mean flow redistribution in curved open channels. *Water Resources Research*, 39(12), 1–14. <https://doi.org/10.1029/2003WR002068>
- Bogoni, M., Putti, M., & Lanzoni, S. (2017). Modeling meander morphodynamics over self-formed heterogeneous floodplains. *Water Resources Research*, 53(6), 5137–5157. <https://doi.org/10.1002/2017WR020726>
- Candel, J. H. J., Kleinhans, M. G., Makaske, B., & Wallinga, J. (2021). Predicting river channel pattern based on stream power, bed material and bank strength. *Progress in Physical Geography*, 45(2), 253–278. <https://doi.org/10.1177/0309133320948831>
- Coulthard, T. J. (2005). Effects of vegetation on braided stream pattern and dynamics. *Water Resources Research*, 41(4), 1–9. <https://doi.org/10.1029/2004WR003201>
- Darlington, R. B. (1970). Is kurtosis really “peakedness?” *American Statistician*, 24(2), 19–22. <https://doi.org/10.1080/00031305.1970.10478885>
- Davies, N. S., Gibling, M. R., McMahon, W. J., Slater, B. J., Long, D. G. F., Bashforth, A. R., et al. (2017). Discussion on ‘Tectonic and environmental controls on Palaeozoic fluvial environments: reassessing the impacts of early land plants on sedimentation’ *Journal of the Geological Society, London*, <https://doi.org/10.1144/jgs2016-063>. *Journal of the Geological Society*, 174(5), 947–950. <https://doi.org/10.1144/jgs2017-004>
- Donovan, M., Belmont, P., & Sylvester, Z. (2021). Evaluating the Relationship Between Meander-Bend Curvature, Sediment Supply, and Migration Rates. *Journal of Geophysical Research: Earth Surface*, 126(3), 1–20. <https://doi.org/10.1029/2020JF006058>
- Finotello, A., D’Alpaos, A., Lazarus, E. D., & Lanzoni, S. (2019). High curvatures drive river meandering: COMMENT. *Geology*, 47(10), e485–e485. <https://doi.org/10.1130/G46761C.1>
- Finotello, A., D’Alpaos, A., Bogoni, M., Ghinassi, M., & Lanzoni, S. (2020). Remotely-sensed planform morphologies reveal fluvial and tidal nature of meandering channels. *Scientific Reports*, 10(1), 1–13. <https://doi.org/10.1038/s41598-019-56992-w>
- Frascati, A., & Lanzoni, S. (2009). Morphodynamic regime and long-term evolution of meandering rivers. *Journal of Geophysical Research: Earth Surface*, 114(2), 1–12. <https://doi.org/10.1029/2008JF001101>
- Frasson, R. P. de M., Pavelsky, T. M., Fonstad, M. A., Durand, M. T., Allen, G. H., Schumann, G., et al. (2019). Global Relationships Between River Width, Slope, Catchment Area, Meander Wavelength, Sinuosity, and Discharge. *Geophysical Research Letters*, 46(6), 3252–3262. <https://doi.org/10.1029/2019GL082027>
- Furbish, D. J. (1988). River-bend curvature and migration: how are they related? *Geology*, 16(8),

- 752–755. [https://doi.org/10.1130/0091-7613\(1988\)016<0752:RBCAMH>2.3.CO;2](https://doi.org/10.1130/0091-7613(1988)016<0752:RBCAMH>2.3.CO;2)
- Guo, X., Chen, D., & Parker, G. (2019). Flow directionality of pristine meandering rivers is embedded in the skewing of high-amplitude bends and neck cutoffs. *Proceedings of the National Academy of Sciences of the United States of America*, *116*(47), 23448–23454. <https://doi.org/10.1073/pnas.1910874116>
- Hickin, E. J., & Nanson, G. C. (1975). The Character of Channel Migration on the Beatton River, Northeast British Columbia, Canada. *Geological Society of America Bulletin*, *86*(4), 487. [https://doi.org/10.1130/0016-7606\(1975\)86<487:TCOCMO>2.0.CO;2](https://doi.org/10.1130/0016-7606(1975)86<487:TCOCMO>2.0.CO;2)
- Hooke, J. M. (2013). River Meandering. In E. Wohl & Schroder (Eds.), *Treatise on Geomorphology* (Vol. 9, pp. 260–288). Washington: Elsevier. <https://doi.org/10.1016/B978-0-12-374739-6.00241-4>
- Hooke, R. L. B. (1975). Distribution of Sediment Transport and Shear Stress in a Meander Bend. *The Journal of Geology*, *83*(5), 543–565. <https://doi.org/10.1086/628140>
- Howard, A. D., & Hemberger, A. T. (1991). Multivariate characterization of meandering. *Geomorphology*, *4*(3–4), 161–186. [https://doi.org/10.1016/0169-555X\(91\)90002-R](https://doi.org/10.1016/0169-555X(91)90002-R)
- Ielpi, A., Lapôtre, M. G. A., Finotello, A., & Ghinassi, M. (2021). Planform-asymmetry and backwater effects on river-cutoff kinematics and clustering. *Earth Surface Processes and Landforms*, *46*(2), 357–370. <https://doi.org/10.1002/esp.5029>
- Ielpi, A., Lapôtre, M. G. A. A., Gibling, M. R., & Boyce, C. K. (2022). The impact of vegetation on meandering rivers. *Nature Reviews Earth and Environment*, *3*(3), 165–178. <https://doi.org/10.1038/s43017-021-00249-6>
- Jerolmack, D. J., & Mohrig, D. (2007). Conditions for branching in depositional rivers. *Geology*, *35*(5), 463. <https://doi.org/10.1130/G23308A.1>
- Lapôtre, M. G. A., Ielpi, A., Lamb, M. P., Williams, R. M. E., & Knoll, A. H. (2019). Model for the Formation of Single-Thread Rivers in Barren Landscapes and Implications for Pre-Silurian and Martian Fluvial Deposits. *Journal of Geophysical Research: Earth Surface*, *124*(12), 2757–2777. <https://doi.org/10.1029/2019JF005156>
- Lazzarin, T., & Viero, D. Pietro. (2023). Curvature-induced secondary flow in 2D depth-averaged hydro-morphodynamic models: An assessment of different approaches and key factors. *Advances in Water Resources*, *171*(November 2022), 104355. <https://doi.org/10.1016/j.advwatres.2022.104355>
- Li, Y., & Limaye, A. B. (2022). Testing Predictions for Migration of Meandering Rivers: Fit for a Curvature-Based Model Depends on Streamwise Location and Timescale. *Journal of Geophysical Research: Earth Surface*, *127*(12), 1–22. <https://doi.org/10.1029/2022JF006776>
- Lightbody, A. F., Kui, L., Stella, J. C., Skorko, K. W., Bywater-Reyes, S., & Wilcox, A. C. (2019). Riparian vegetation and sediment supply regulate the morphodynamic response of an experimental stream to floods. *Frontiers in Environmental Science*, *7*(APR), 1–14. <https://doi.org/10.3389/fenvs.2019.00040>
- Moors, J. J. A. (1986). The Meaning of Kurtosis: Darlington Reexamined. *The American Statistician*, *40*(4), 283. <https://doi.org/10.2307/2684603>
- Ottevanger, W., Blanckaert, K., & Uijtewaal, W. S. J. (2012). Processes governing the flow redistribution in sharp river bends. *Geomorphology*, *163–164*, 45–55. <https://doi.org/10.1016/j.geomorph.2011.04.049>
- Ruppert, D. (1987). What is kurtosis? An influence function approach. *American Statistician*, *41*(1), 1–5. <https://doi.org/10.1080/00031305.1987.10475431>
- Santos, M. G. M., Mountney, N. P., Peakall, J., Thomas, R. E., Wignall, P. B., & Hodgson, D. M. (2017). Reply to Discussion on ‘Tectonic and environmental controls on Palaeozoic fluvial environments: reassessing the impacts of early land plants on sedimentation’ *Journal of the Geological Society*, London, <https://doi.org/10.1144/jgs2016-063>. *Journal of the Geological Society*, *174*(5), 950–952. <https://doi.org/10.1144/jgs2017-031>
- Santos, M. G. M., Mountney, N. P., & Peakall, J. (2017). Tectonic and environmental controls on palaeozoic fluvial environments: Reassessing the impacts of early land plants on sedimentation. *Journal of the Geological Society*, *174*(3), 393–404.

<https://doi.org/10.1144/jgs2016-063>

- Santos, M. G. M., Hartley, A. J., Mountney, N. P., Peakall, J., Owen, A., Merino, E. R., & Assine, M. L. (2019). Meandering rivers in modern desert basins: Implications for channel planform controls and prevegetation rivers. *Sedimentary Geology*, 385, 1–14. <https://doi.org/10.1016/j.sedgeo.2019.03.011>
- Schwenk, J., & Fofoula-Georgiou, E. (2016). Meander cutoffs nonlocally accelerate upstream and downstream migration and channel widening. *Geophysical Research Letters*, 43(24), 12,437–12,445. <https://doi.org/10.1002/2016GL071670>
- Seminara, G. (2006). Meanders. *Journal of Fluid Mechanics*, 554(1), 271–297. <https://doi.org/10.1017/S0022112006008925>
- Stanislowski, L. V., Kronenfeld, B. J., Battenfield, B. P., & Shavers, E. J. (2023). At what scales does a river meander? Scale-specific sinuosity (S3) metric for quantifying stream meander size distribution. *Geomorphology*, 436(May), 108734. <https://doi.org/10.1016/j.geomorph.2023.108734>
- Sylvester, Z., Durkin, P., Covault, J. A., & Sharman, G. R. (2019a). High curvatures drive river meandering: REPLY. *Geology*, 47(10), e486–e486. <https://doi.org/10.1130/G46838Y.1>
- Sylvester, Z., Durkin, P., Covault, J. A., & Sharman, G. R. (2019b). High curvatures drive river meandering. *Geology*, 47(10), e486–e486. <https://doi.org/10.1130/G46838Y.1>
- Tal, M., & Paola, C. (2007). Dynamic single-thread channels maintained by the interaction of flow and vegetation. *Geology*, 35(4), 347–350. <https://doi.org/10.1130/G23260A.1>
- Török, G. T., & Parker, G. (2022). The influence of riparian woody vegetation on bankfull alluvial river morphodynamics. *Scientific Reports*, 12(1), 1–15. <https://doi.org/10.1038/s41598-022-22846-1>
- Zhiqiang, L., Jianming, W., Junyu, Z., Haitao, L., Baoqing, L., Jie, S., & Chunlei, Z. (2008). The statistical meaning of kurtosis and its new application to identification of persons based on seismic signals. *Sensors*, 8(8), 5106–5119. <https://doi.org/10.3390/s8085106>

REVIEWERS' COMMENTS

Reviewer #2 (Remarks to the Author):

The authors have addressed all the issues I raised with sufficient detail. Thank you for taking the time to respond to all the comments.

Reviewers #3 and #4 – Co-Review (Remarks to the Author):

We enjoyed re-reading the revised manuscript. In short, the authors have done a commendable job at addressing the reviewers' comments and we believe that the manuscript will be a broadly-read contribution. The authors restructured the paper, to present all the results before going into an expanded discussion on the morphometrics and dynamics side-by-side (in the Discussion and Implications) section. The current structure works well, and is a marked improvement from the previous version. We especially liked the combined nuance discussion on the complex connections between cutoffs, avulsions, and bend maturity, and its relation to migration patterns. The authors better address data aggregation in the main text and the figures, and the color contrast significantly help figure readability in all cases.

One last comment on lines 158-161 in revised manuscript – are the authors referring to the statistical difference in migration rates shown in figure 4a? If so, it seems like the P value might be reported for the bend-scale data as opposed to reach-scale data. It seems like the reach-scale data show significant overlap. Could the authors double check this in the final version? Visually, the data do not seem so far off from each other to yield the small P value reported unless the differences are being swamped by the log scale.

In summary, we believe that our concerns have been addressed in the revisions and we look forward to the manuscript in publication. This manuscript was co-reviewed as part of the Nature Communications initiative to facilitate training in peer review and to provide appropriate recognition for Early Career Researchers.

RESPONSE TO REVIEWERS ON

"Vegetation enhances curvature-driven dynamics in meandering rivers" by Finotello et al.
(Manuscript ID: NCOMMS-23-37093A)

Please note that in this response, *italics* refer to the text of the reviewers' comments, our detailed replies are in black, and the new text of the revised version is in **bold** font. Line numbers in our replies refer to the revised version of the manuscript.

REVIEWER #2 (REMARKS TO THE AUTHOR):

The authors have addressed all the issues I raised with sufficient detail. Thank you for taking the time to respond to all the comments.

Thank you for this supportive assessment of our work.

REVIEWER #3 AND #4 - CO-REVIEW (REMARKS TO THE AUTHOR):

We enjoyed re-reading the revised manuscript. In short, the authors have done a commendable job at addressing the reviewers' comments and we believe that the manuscript will be a broadly-read contribution. The authors restructured the paper, to present all the results before going into an expanded discussion on the morphometrics and dynamics side-by-side (in the Discussion and Implications) section. The current structure works well, and is a marked improvement from the previous version. We especially liked the combined nuance discussion on the complex connections between cutoffs, avulsions, and bend maturity, and its relation to migration patterns. The authors better address data aggregation in the main text and the figures, and the color contrast significantly help figure readability in all cases.

One last comment on lines 158-161 in revised manuscript – are the authors referring to the statistical difference in migration rates shown in figure 4a? If so, it seems like the P value might be reported for the bend-scale data as opposed to reach-scale data. It seems like the reach-scale data show significant overlap. Could the authors double check this in the final version? Visually, the data do not seem so far off from each other to yield the small P value reported unless the differences are being swamped by the log scale.

In summary, we believe that our concerns have been addressed in the revisions and we look forward to the manuscript in publication. This manuscript was co-reviewed as part of the Nature Communications initiative to facilitate training in peer review and to provide appropriate recognition for Early Career Researchers.

Thank you for this supportive assessment of our work.

Regarding data reported in Figure 4a, we have double-checked them and can confirm the accuracy of the reported values, with the logscale helping to minimize visual differences between the three sets of data.